# Has Third-Party Monitoring Improved Water Pollution Data Quality? Evidence from National Surface Water Assessment Sections in China

**Liange Zhao, Mengyao Hong and Xueyuan Wang ***

School of Economics, Zhejiang Gongshang University, Hangzhou 310018, China; hhyzlg@163.com (L.Z.); ncj2800@dingtalk.com (M.H.)
* Correspondence: wxyzyx@zjgsu.edu.cn; Tel.: +86-13588493453

**Abstract:** In China, the central government assesses local governments based on data monitored and reported by local agencies, and the accuracy of local statistics has been controversial. In order to further guarantee the authenticity and reliability of surface water monitoring data, the central government will gradually withdraw the local monitoring powers of the national surface water assessment section and implement third-party monitoring to achieve "national assessment and national monitoring." This paper is based on the time-point water data of important national water quality automatic monitoring stations from 2015 to 2020, using the McCrary (2008) density test to infer possible data manipulation phenomena, and analyze whether third-party monitoring has improved the accuracy of China's environmental data. The results of the study show that between 2015 and 2020, the observed 81 monitoring sites had varying degrees of data discontinuity. The discontinuity of the data after third-party monitoring was reduced in dissolved oxygen (DO) measurement, an important indicator in the assessment, implying that third-party monitoring has improved the quality of water environment data and the accuracy of the data. The research in this article provides a reference for third-party participation in environmental governance and proves that the participation of these organizations can reduce data manipulation behaviors of local governments and ensure the effectiveness of environmental data.

**Keywords:** surface water quality monitoring; third-party monitoring; water environment management; incentive compatibility; data manipulation

## 1. Introduction

Environmental monitoring is an important foundation for environmental protection. It is extremely important to ensure the normal operation of monitoring equipment and the authenticity of monitoring data. Research on water pollution supervision shows that environmental monitoring and law enforcement activities are effective in improving water quality [1]. Although the newly revised "Water Pollution Prevention and Control Law of the People's Republic of China" has added regulations on the authenticity and accuracy of monitoring data, the falsification of water pollution detection data still occurs frequently in various places. Comparing official data with original data shows that there is the possibility of data manipulation in data distribution near the boundary of the "blue sky day" [2]. The improvement in Beijing's air quality during the 2008 Olympic Games was real, but only temporary [3]. Ghanem and Zhang [4] used regression discontinuity testing to find that about 50% of the cities reported $PM_{10}$ indices with obvious discontinuities at the "blue sky day" cut-off point. Much air pollution data manipulation exists in various cities every year, as described by Ghanem et al. [5].

Most developing countries' water resource management lacks efficiency, and there are obvious shortcomings in the fields of data collection, analysis and publication, and resource planning [6]. In China, many policies are formulated based on data reported by lo-

cal governments, and statistical data is linked to local government performance appraisals and possible future promotions [7], which provides motivation for data fraud. Taking advantage of information asymmetry between the central and local governments, local officials may exaggerate economic achievements and underestimate environmental pollution [4]. In the past, the process of publicizing water quality information mainly consisted of local monitoring departments collecting statistical information and reporting it to the central government, which lacked supervision by third-party organizations. Third-party monitoring is actually a manifestation of environmental information disclosure. The joint participation of third-party monitoring in environmental management can improve the accuracy of data and increase the transparency of information. Many studies have shown that environmental information disclosure is beneficial to environmental governance. Evidence in the literature shows that TRI Plan (Environmental Information Disclosure Plan implemented in the United States in 1986) is beneficial to environmental governance [8,9]. Based on plant-level data from the United States and Indonesia, Pargal et al. [10] verified that disclosure of environmental information has a significant positive impact on reducing pollutant emissions. Kathuria [11] proposed that environmental information disclosure plays a positive role in reducing water pollution in India. In order to improve the government's environmental monitoring capabilities and protect the public's right to know, third-party monitoring has been implemented since October 2017, and the model of "automatic monitoring as the mainstay and manual monitoring as the supplement" has been fully promoted. Third-party monitoring means that the collection and analysis of water samples of the national assessment section are assigned to different units, allowing third-party agencies to participate in the collection of environmental data at the local level, strengthening the cooperation of the central government with market forces, rather than rely solely on local monitoring department to collect statistical data. The participation of third-party institutions in environmental governance has played an important role in external independent supervision, enabling the central government to circumvent the lies of local governments [12]. Although government agencies have demonstrated the importance of third-party environmental monitoring [13], there is little substantive and systematic analysis in the literature to study the impact of such major policy innovations on data quality, and the impact of changes on environmental governance decisions in China [14]. Due to the relatively easy access of air quality index data and the relatively clear and single assessment index, many scholars choose to study air quality index. Most of the existing studies on environmental data manipulation are also based in this field, while there are few articles on water environment data manipulation.

Through empirical research, this paper uses the McCrary (2008) density test method to analyze the effectiveness of third-party monitoring in improving the quality of water environment data based on the real-time monitoring data of the National Water Station released on the Internet by the China Environmental Monitoring Center. Compared with the existing literature, the innovations of this article are the following. (1) Discussion of the impact of third-party monitoring on water pollution data quality from the perspective of incentive compatibility theory, linking the phenomenon of data manipulation with the theory of incentive compatibility, and analyzing the principal-agent crisis in current environmental governance. (2) In terms of content, from the point of water pollution data reporting, this article uses real-time water pollution data released by the China Environmental Monitoring Station. Compared with the air quality index, the water quality assessment index and supervision process are more complex, and there are few articles that include testing water pollution data. (3) In terms of research methods, this paper uses the McCrary (2008) density test to inspect the water pollution data, effectively avoiding the endogenous problem, and then launches a robustness test to make the research results more reliable.

The rest of this paper is organized as follows. Section 2 describes the research background and mechanism analysis. Section 3 summarizes the measurement model setting

and variable explanation. Section 4 is the empirical result analysis, and the final section presents the research conclusion and policy implications.

## 2. Background and Mechanism Analysis

### 2.1. Background

In September 2017, the Ministry of Environmental Protection issued the "Implementation Plan for the Separation of Sampling and Measurement of National Surface Water Environmental Quality Monitoring Network". Since October 2017, a total of 2050 national assessment sections have been included in the third-party monitoring of national surface water, with a frequency of once a month. Third-party monitoring means that the work of collecting and analyzing water samples of the national assessment section is assigned to different units, changing the existing local monitoring mode and delinking from the stakeholders in the mechanism. The specific technical route of third-party monitoring is shown in Figure 1. The China Environmental Monitoring Station makes a unified implementation plan. The third-party organization samples according to unified technical specifications, and the water samples are encrypted and randomly distributed to each analysis station. The original monitoring data is directly transmitted to the central monitoring station, and the quality control of all links in the whole process is emphasized, which can ensure the truth and accuracy of the data to a greater extent. October to December 2017 was the trial period of third-party monitoring, with 2050 test sections, of which 1854 sections were subject to third-party monitoring and 196 sections were subject to territorial monitoring. The local monitoring stations that originally undertook the monitoring tasks of the 1854 national examination sections carried out simultaneous monitoring during the trial operation of the test operation.

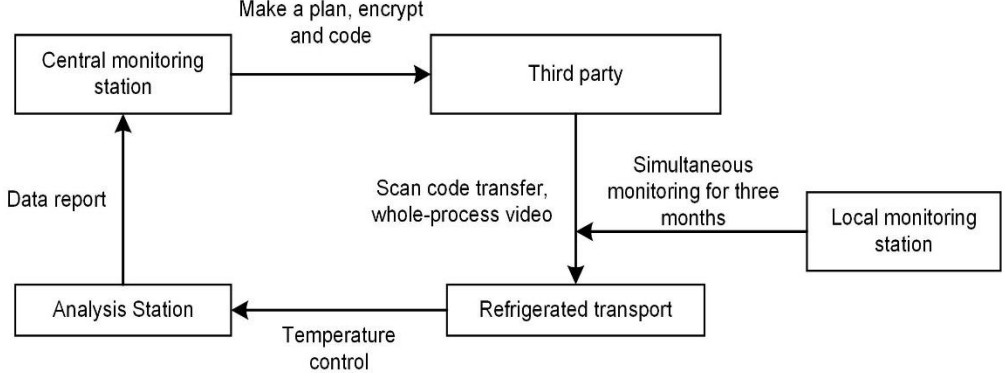

**Figure 1.** Roadmap of third-party monitoring technology. Note: The picture is referenced from "Implementation Plan for the Separation of Sampling and Measurement of National Surface Water Environmental Quality Monitoring Network" http://www.cnemc.cn/, accessed on 8 July 2020.

Environmental monitoring construction is the core task of the country's environmental supervision capacity building. The government's investment in this field has been increasing year by year, and tremendous results have been achieved. In recent years, the work of water quality environmental monitoring has also been improved. In terms of the construction of the national surface water monitoring station network, since 2016, the Monitoring Department, together with the Central monitoring station, has tried various ways to withdraw the power of national surface water environmental quality monitoring. Table 1 analyzes the advantages and disadvantages of the various monitoring schemes that have been tried. Finally, through the comparison of various results and taking full consideration of the actual status quo, the mode of "automatic monitoring as primary and manual monitoring as secondary" was selected.

**Table 1.** The advantages and disadvantages of the national surface water monitoring program.

| Monitoring Program | Advantage | Disadvantage |
|---|---|---|
| Joint monitoring | Reduce data disputes. | Only applicable to cross-border sections, high cost, no rules for data identification. |
| Third-party simultaneous monitoring | Wide application, easy to implement. | Poor analysis and testing level, some indicators are not standardized. |
| Remote quality control | Experts recode samples, effective quality control. | Many experts are required, and the cost is extremely high. |
| Automatic station monitoring | It can continuously monitor the actual water body in unattended state. Dissolved oxygen (DO), permanganate index ($COD_{Mn}$), ammonia nitrogen ($NH_3$-N), total phosphorus is consistent and comparable with manual data. | |
| Separation of Sampling and Measurement | Break the territorial monitoring model, avoid administrative intervention; give full play to the role of local stations and social institutions, and the cost is reasonable. | |

Note: The table is referenced from "Implementation Plan for the Separation of Sampling and Measurement of National Surface Water Environmental Quality Monitoring Network", http://www.cnemc.cn/, accessed on 8 July 2020.

*2.2. Mechanism Analysis*

Holmstrom and Milgrom [15] proposed that the central government entrusts local governments to manage multiple affairs in their jurisdictions, and avoid conflicts among various tasks. Local officials would choose to increase their efforts on one task, which would lead to poor performance of the other task. At this time, the setting of incentives became the key to coordinating tasks. In order to ultimately realize personal self-interest and social interests, incentive compatibility theory requires the participants' personal interests and the designer's established goals to reach an agreement [16]. When local officials strive to achieve their political goals, the allocation of environmental resources inevitably affect the government's decision-making on economic development [17]. Environmental protection targets are binding targets designed to prevent the most serious situation, and if officials cannot obtain a relatively high position in the competition for economic growth they have no incentive to achieve the binding targets [18]. Although the development goals of the central government are specifically specified in each specific period, the overall goal of the central government is to achieve equilibrium [19].

Local monitoring stations face the task requirements of different subjects. In addition to accepting the task control of the higher-level monitoring station in terms of environmental tasks, there is also task pressure from the local government, which is likely to result in the conflict of multiple objectives. Local environmental protection bureaus and their subordinate departments are responsible for environmental monitoring [20]. However, these institutions are often at the bottom of the political system, their power and status are very limited, and they have little deterrent effect on data manipulation [13]. Jingdong [21] proposed that the project system is the core forming a hierarchical governance mechanism between the central and local governments, which has produced many unexpected consequences for the grassroots society. Under the hierarchical responsibility system, statistics bureaus at all levels are responsible for the statistical work at their own levels, and local governments rather than the central government have the right to manage the statistics bureaus at that level [12]. Evidence in the literature shows that the promotion probability of officials is closely related to economic performance [22,23]. Local governments may sacrifice local environmental resources for the economic development of the region. When conflicts arise between various tasks, the local government that has the most comprehensive and accurate local information has a great incentive to tamper with relevant data.

A systematic study of data manipulation originated in the United States in the 1950s. In this information society, we are exposed to more and more data, such as economic data, environmental data, and energy data and so on. These data are often disseminated

through "packaging". Data manipulation is a common phenomenon rooted in various interests. Zhang et al. [24] used McCrary's (2008) density test and found that in order to obtain subsidies from the Granary County Subsidy Program (GCSP), counties below the threshold had an incentive to over-report their grain production output. The study of Firpo et al. [25] found that individuals manipulated their income by voluntarily reducing the labor supply, thereby making them eligible to participate in the family grant program. P. Zhang et al. [26] used satellite night lighting data to correct the GDP growth rate, and found that the reduction in energy intensity was overestimated due to inaccurate GDP data.

A third party as a vested interest and may produce unreliable results. For example, in many regulated markets, private third-party auditors are selected and paid for by the company itself, and may underreport factory emissions [27]. Vidovic et al. [28] used facility-level panel data from factories in the United States from 1996 to 2010 and found that the third party had no significant impact on voluntary emission reductions. However, some studies have shown that the introduction of a third party can improve the efficiency of environmental supervision. Niu et al. [14] found that third-party environmental monitoring can improve the accuracy of China's environmental data. Zhou et al. [29] proposed that sample cities that adopt a third-party governance model can more effectively improve environmental pollution. The introduction of a third-party evaluation and public supervision system can balance the contradiction between economic growth and environmental pollution [30]. Although third-party organizations are of great significance to environmental regulation, little is known about how they improve the accuracy of data and their impact on environmental governance

Suppose that the behavior of local government is divided into two types: data manipulation and public data. $U_1$ is the reward for local government data manipulation, $U_2$ is the reward for local government data disclosure, and b is the probability of local government data manipulation. Figure 2 shows the environmental supervision organization system after third-party monitoring. The state unifies the sampling time and technical methods and assigns a third-party monitoring agency to take charge of random sampling. The samples are encrypted and sent to the analysis station for analysis. This breaks the original territorial monitoring model, cut off the connection with the local government, and avoids administrative intervention. By implementing third-party monitoring, the central government hopes to improve and maintain the necessary environmental monitoring capabilities and reduce data manipulation by local governments in order to collect high-quality data required for decision-making. It helps the public to better obtain water quality-related information and protects the public's right to know the environment.

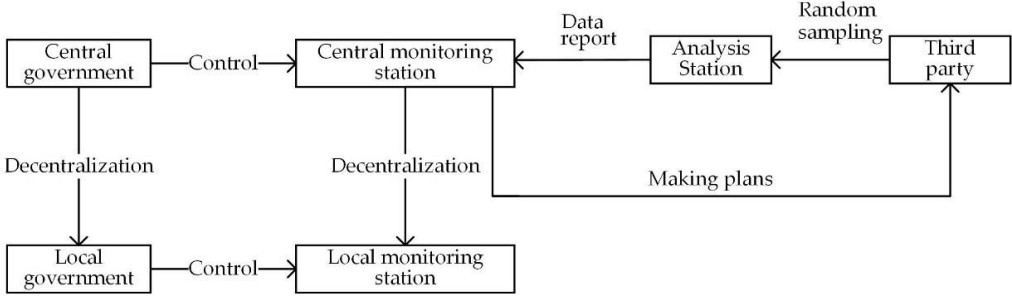

**Figure 2.** Environmental supervision organization system after third-party monitoring.

Since the monitoring frequency of the third-party supervision is once a month, it is impossible to fully expose the data manipulation behavior of the local government. It is assumed that the probability of the third-party organization finding abnormal data is $\rho$. If the local government is found to have data manipulation, the central government punishes the local government as $P_L$. Next, we analyze the impact of third-party monitoring on the probability of local government data manipulation. If the local government chooses data

manipulation, the effect of data manipulation is not lower than the effect of disclosing the data, that is:

$$b(U_1 - \rho P_L) \geq (1 - b)U_2 \tag{1}$$

Simplifying the above formula we get:

$$U_1 \geq (1/b - 1)U_2 + \rho P_L \tag{2}$$

Here, $b$ is the probability of data manipulation by the local government. The greater the values of b, the greater the probability that the local government will choose data manipulation. As $b$ increases, $U_1$ decreases. This shows that after the third-party monitoring, if the local government still chooses to manipulate the data, its reward will decrease, thereby inhibiting the local government's data manipulation behavior and making it present the real water quality. Based on this, this paper proposes the following research hypothesis: third-party monitoring will improve the quality of water pollution data and improve the environmental monitoring capability of the government, so that the probability of data manipulation by local governments will be reduced.

## 3. Models and Date

### 3.1. Methods

To check the accuracy of the official data, the best way is to use independent data sources to compare with official data. However, independent data used for comparative analysis with official data is often difficult to obtain. In the absence of data manipulation, the concentration distribution of various indicators of surface water pollution should be a continuous or smooth curve. Local officials are most likely to cheat at the critical point of the classification standard when they need to manipulate the data through various motivations. If the data values are decreased to a value slightly higher than the target level standard limit and the water quality level is reduced, or increased to values slightly lower than the target standard limit and the water quality level increased, the changes would be small and difficult to notice. If such a situation occurs repeatedly, this indicates that there is a suspicion of data manipulation at the particular station. Therefore, this article uses the McCrary (2008) density test [31] to examine possible data manipulation in National Surface Water Sections. One disadvantage of the method is that if a certain site is manipulated by subtracting a fixed number from the pollutant concentration, this will not cause discontinuities, but just an average deviation of the distribution. In order to be able to operate without causing a discontinuity at the cut-off, the local government must know the distribution of water quality throughout the period. However, water quality monitoring sites must report their data daily, so local governments are unlikely to manipulate the data without leading to a discontinuity at the cut-off [4].

If the individual knows the grouping rules in advance and can choose to enter the left or right side of the breakpoint through their own efforts, it leads to uneven distribution on the left and right sides of the breakpoint, and the left and right limits will be different. The method of McCrary (2008) examines whether the density function of the grouping variable is continuous at the cut-off and test the null hypothesis:

$$H_0: \hat{\theta} = \ln \lim_{x \downarrow c} f(x) - \ln \lim_{x \uparrow c} f(x) = \ln \hat{f}^+ - \ln \hat{f}^- = 0 \tag{3}$$

The first step is to divide the grouping variables into equal distances as much as possible on both sides of the cut-off $c$, draw a very rough histogram, set the bin size to $b$, and record the center position of each group as the variable $X_j = \left\{ \ldots, c, c - \frac{3b}{2}, c - \frac{b}{2}, c + \frac{b}{2}, c + \frac{3b}{2}, \ldots \right\}$. Then the standardized frequency $Y_j$ of each group is calculated, that is, the frequency divided by $nb$ (n is the sample size). The second step is to use a triangular kernel to perform a local linear regression of $Y_j$ against $X_j$. For the value of the grouping variable $r_0 = \{ \ldots, c - 2b, c - b, c + b, c + 2b, \ldots \}$, the estimated value of the density function $\hat{f}(r_0)$ and the

standard error $SE\left[\left(\hat{f}(\mathbf{r}_0)\right)\right]$ can be obtained. Finally, by calculating the estimated value of $\theta$ and its standard error, we can check whether the density function $f(\mathbf{x})$ is continuous at $x = c$. The function estimate is:

$$\hat{\theta} = \ln \lim_{x \downarrow c} f(\mathbf{x}) - \ln \lim_{x \uparrow c} f(\mathbf{x}) = \ln \hat{f}^+ - \ln \hat{f}^-$$

$$= \ln \left\{ \sum_{X_j > c} k\left(\frac{X_j - c}{h}\right) \frac{S_{n,2}^+ - S_{n,1}^+ (X_j - c)}{S_{n,2}^+ S_{n,0}^+ - S_{n,1}^+ S_{n,1}^+} Y_j \right\} - \ln \left\{ \sum_{X_j < c} k\left(\frac{X_j - c}{h}\right) \frac{S_{n,2}^- - S_{n,1}^- (X_j - c)}{S_{n,2}^- S_{n,0}^- - S_{n,1}^- S_{n,1}^-} Y_j \right\} \tag{4}$$

Among them:

$$S_{n,k}^+ = \sum_{X_j > c} K\{(X_j - c)/\mathrm{h}\}(X_j - c)^k, \tag{5}$$

$$S_{n,k}^- = \sum_{X_j < c} K\{(X_j - c)/\mathrm{h}\}(X_j - c)^k, \tag{6}$$

$$\mathrm{K}(\mathrm{t}) = \max\{0, 1 - |t|\}, \tag{7}$$

In the absence of data manipulation, the distribution of various index values of surface water quality should be a continuous or smooth curve. In this case, the null hypothesis $H_0$ is accepted, and there is no significant difference between the left and right limits of the cut-off. When there are significant jumps on the left and right sides of the cut-off c, the null hypothesis can be rejected at a certain level of significance, and there is a possibility of data manipulation.

Specifically, we compared the changes in the discontinuity of water environment data before and after third-party monitoring. If the water environment data is manipulated (for example, under-reported or over-reported), the left limit will not be equal to the right limit at the cut-off. We compared the changes in the discontinuity of the data before and after third-party monitoring to test whether the accuracy of the data improved.

*3.2. Date Sources and Index Design*

The data used in the empirical test part comes from the China Environmental Monitoring Center. The website publishes the national real-time monitoring data of surface water quality and provides real-time monitoring data query, including pH, dissolved oxygen (DO), permanganate ($COD_{Mn}$), ammonia nitrogen ($NH_3$-N) and total organic carbon (TOC). Each site can provides six monitoring results for each monitoring item every day, at a frequency of 4 h, at 0:00, 4:00, 8:00, 12:00, 16:00, 20:00 and 24:00. Automatic water quality monitoring stations often suspend operations due to various special weather conditions or technical reasons during their daily operation. Newly built automatic water quality stations are also put into use every year, so the data records of each station are not completely continuous. To analyze the changes in the accuracy of the data before and after third-party monitoring, our research selected sites that have continuous records from 2015 to 2020. After preliminary screening and processing, the final data used in this article comes from a total of 81 national automatic monitoring sites for surface water quality in 31 provinces, autonomous regions, and municipalities across the country. The China Environmental Monitoring Station is responsible for the business management of each station, and the daily operation and maintenance work is entrusted to the local environmental monitoring station.

The time-point monitoring items announced by the China Environmental Monitoring Center mainly include five indicators: dissolved oxygen (DO), permanganate index ($COD_{Mn}$), ammonia nitrogen ($NH_3$-N), pH value and total organic carbon (TOC): see details in Table 2 below. In order to make the expression of water quality more intuitive and direct, according to the "Surface Water Environmental Quality Standard" (GB3838-2002), some item values can be one-to-one corresponding to water quality categories. The specific limits are shown in Table 3. Water quality can be divided into five categories according to

each index value. The higher the water quality category, the higher the pollution level, and the worse the water quality.

**Table 2.** The meaning of the main monitoring indicators.

| Name of Index | Meaning |
|---|---|
| Dissolved oxygen (DO) | Represents molecular oxygen dissolved in water. The dissolved oxygen index in water is one of the important indicators reflecting the quality of water bodies. Surface water that contains organic pollutants has reduced dissolved oxygen when the organic pollutants decompose under the action of bacteria, making the water body black and smelly, and causing fish, shrimp and other aquatic organisms to die. In natural water with good fluidity (good exchange with air), the saturated concentration of dissolved oxygen is related to temperature and air pressure. At zero degrees, the saturated oxygen content in water is 14.6 mg/L, and at 25 °C it is 8.25 mg/L. When algae grow in water bodies, oxygen is generated due to photosynthesis, which causes the surface dissolved oxygen to rise abnormally and exceed the saturation value. |
| Permanganate Index ($COD_{Mn}$) | Using potassium permanganate as the oxidant, the amount consumed when processing surface water samples is expressed in mg/L of oxygen. Under these conditions, both reducing inorganic substances (ferrous salts, sulfides, etc.) and organic pollutants in the water can consume potassium permanganate, which is often used as a comprehensive indicator of the degree of surface water pollution by organic pollutants. The potassium permanganate method, also known as chemical oxygen demand, is different from the chemical oxygen demand (COD) of the potassium dichromate method, which is often used for wastewater discharge monitoring. |
| Ammonia nitrogen ($NH_3$-N) | Ammonia nitrogen exists in water in the form of molecular ammonia in the dissolved state (also known as free ammonia, $NH_3$) and in the form of ammonium salt ($NH_4+$). The ratio of the two depends on the pH value and temperature of the water. The ammonia nitrogen is expressed by the amount of N element content. The sources of ammonia nitrogen in water are mainly domestic sewage, industrial wastewater and surface runoff (mainly fertilizer used in farmland enters rivers, lakes and reservoirs through surface runoff). |
| pH value (pH) | An indicator that characterizes the acidity and alkalinity of water. A pH value of 7 is indicated as neutral, a value less than 7 is acidic, and a value greater than 7 is alkaline. The pH value of natural surface water is generally between 6 and 9. When algae grow in the water body, the pH value of the surface increases due to the absorption of carbon dioxide by photosynthesis. |
| Total organic carbon (TOC) | Another comprehensive index representing the content of organic matter in water bodies. When organic matter in the water sample is combusted, by measuring the carbon dioxide ($CO_2$ generated the total organic carbon content can be expressed in terms of the amount of the C element. For water samples with the same chemical composition, there is a correlation between total organic carbon and the permanganate index. |

Source: China Environmental Monitoring Center, http://www.cnemc.cn, accessed on 8 July 2020.

**Table 3.** Standard limits of basic items of surface water quality.

| Index | Class I | Class II | Class III | Class IV | Class V |
|---|---|---|---|---|---|
| PH | | | 6–9 | | |
| DO | ≥7.5 | ≥6 | ≥5 | ≥3 | ≥2 |
| $COD_{Mn}$ | ≤2 | ≤4 | ≤6 | ≤10 | ≤15 |
| $NH_3$-N | ≤0.15 | ≤0.5 | ≤1.0 | ≤1.5 | ≤2.0 |

Source: China Environmental Monitoring Center, http://www.cnemc.cn, accessed on 8 July 2020.

There is currently no evaluation standard for total organic carbon (TOC), and there are many missing values in the data. The indicator pH is dimensionless. The pH value of natural surface water is generally 6–9. There is no specific standard for pH value of the five types of water quality. It can be concluded from Table 3 that the higher the permanganate index ($COD_{Mn}$) and ammonia nitrogen ($NH_3$-N) content, the more serious the water pollution and the higher the water quality category. The opposite is true for dissolved oxygen (DO) in that the higher its content, the lower the water quality category and the better the water quality. Considering the actual distribution of the data and the main idea of the article, the three main pollution indices of dissolved oxygen (DO), permanganate



index ($COD_{Mn}$) and ammonia nitrogen ($NH_3$-N) were be selected and tested at the critical points of each water quality index.

As introduced above, in this part, the six daily real-time water quality data from 2015–2020 released by the China Environmental Monitoring Station are used, including the three indicators, dissolved oxygen (DO), permanganate ($COD_{Mn}$) and ammonia nitrogen ($NH_3$-N). We deleted records of individual reporting months with integer values for each year to ensure that the reporting of indicators had the smallest scale unit accurate to 0.01. October to December 2017 was the trial period of third-party monitoring. Data were collected using an automatic monitoring system, and actual data distribution assessed with reference to the book "Introduction to the Automatic Monitoring System for Surface Water Quality." Some unreasonable extreme values were deleted during the data sorting process. The total sample size of each indicator exceeded 500,000. Table 4 reports the descriptive statistics of the samples before and after third-party monitoring. According to the mean values of dissolved oxygen (DO) at each observation site, water quality could be classified as Class I. From the mean value of permanganate ($COD_{Mn}$) and ammonia nitrogen ($NH_3$-N), the average water quality grade could be classified as Class II.

**Table 4.** Descriptive statistics of the sample.

| Variable | Observations | Mean | Std. Dev. | Min | Max |
|---|---|---|---|---|---|
| | Before third-party monitoring | | | | |
| DO | 340,812 | 7.824 | 2.627 | 0.01 | 19 |
| $COD_{Mn}$ | 313,752 | 4.018 | 2.861 | 0.01 | 20 |
| $NH_3$-N | 300,573 | 0.381 | 0.623 | 0.01 | 6 |
| | After third-party monitoring | | | | |
| DO | 391,717 | 8.821 | 2.663 | 0.01 | 20 |
| $COD_{Mn}$ | 366,827 | 3.735 | 2.318 | 0.01 | 20 |
| $NH_3$-N | 368,306 | 0.311 | 0.467 | 0.01 | 6 |
| | Total sample | | | | |
| DO | 732,529 | 8.357 | 2.693 | 0.01 | 20 |
| $COD_{Mn}$ | 680,579 | 3.865 | 2.587 | 0.01 | 20 |
| $NH_3$-N | 668,879 | 0.343 | 0.544 | 0.01 | 6 |

## 4. Results

### 4.1. McCrary Test Results

As shown in Table 3, the higher the values of the permanganate index ($COD_{Mn}$) and the ammonia nitrogen ($NH_3$-N) index, the worse the water quality and the higher the water pollution level. The dissolved oxygen (DO) index is quite special. It is positively related to water quality and inversely related to the water quality grade of surface water evaluation. If there were not enough samples near the cut-off point, a result could not be derived, and we treated this as a missing value. October to December 2017 was the trial period for third-party monitoring. The local monitoring stations that originally undertook the monitoring task carried out synchronous monitoring during the trial operation. The monitoring data for these three months were influenced by local monitoring stations and third-party monitoring was not fully implemented. To test the impact of third-party monitoring on water environment data, data from October to December were deleted during the test. According to the time of execution of third-party monitoring, we used the time-point data of water pollutant concentration from January 2015 to September 2017 and January 2018 to May 2020 and conducted the McCrary test on each site at the classification points of each indicator.

Although a graph is more intuitive, the t-statistic is more accurate because it is obtained by standardizing the variance. We used the t-statistic at the 5% significance level to detect data manipulation behavior [4]. Comparing the t-statistic of the two time periods, it was found that at the statistical significance level of 5%, the data from some sites changed

from discontinuous to continuous, and some sites did not show manipulation behavior before or after third-party monitoring. Some of the station data changed from continuous to discontinuous. This is because the third-party monitoring policy is progressive rather than a one-size-fits-all. Since October 2017, the third-party monitoring mode has been adopted, but the specific time for the implementation of third-party monitoring in each watershed was not the same. Due to long journeys involved in sampling, some sites could not complete sampling within 18 h and could not realize third-party monitoring, so data manipulation still existed in some sites. The following graphs show the running results of three anonymous monitoring points before and after third-party monitoring to illustrate changes in data discontinuity.

In Figure 3, the left figures represent results before third-party monitoring, and the right figures the results after third-party monitoring. Comparing the graphs on the left and right sides, the absolute values of the t-statistic for the results from stations A, B, and C before third-party monitoring are all greater than 3, so the null hypothesis that the density function is continuous at the cut-off is rejected. The confidence intervals of the density function estimates on both sides of the cut-off are not overlapped, and there are significant differences in the density functions on both sides of the classification point. Therefore, there is a possibility of data manipulation at this classification point. After third-party monitoring, the absolute values of the t-statistic for the results of stations A, B, and C were all less than 1.96. The confidence intervals of the estimated density function on both sides of the cut-off have overlapping intervals, and there is no significant difference in the density function on both sides of the classification point. Therefore, the null hypothesis that the density function is continuous at the cut-off is acceptable, indicating that adopting a third-party monitoring policy could reduce data manipulation behavior.

The integrated statistics of the results after the operation found that the 81 sites with a statistical significance level under 5% had more or less different degrees of data discontinuity during the five years from 2015 to 2020. The results show two opposite strategies: one to underreport the level of water pollution to make the water quality better, and the other to over-report the level to make the water quality worse. The diagrams and t-statistics were obtained by testing the data of 81 stations before and after third-party monitoring. The value of the t-statistic was used to judge which sections had data manipulation, and the results were classified and sorted. Detailed results are shown in Table A1 in the Appendix A. For privacy protection of each monitoring point, we use digital codes to indicate the name of the monitoring point.

After combining statistics of the indicators at the same level, it was found that the phenomenon of "underreporting" and "over-reporting" occurred in a higher proportion near the three standard limits of Class I, Class II, and Class III classification points. We first calculated the arithmetic mean value of the absolute value of t-statistics generated by the McCrary test of the three indicators at each grading point, and then drew scatter plots of the t-statistics of the three indicators before and after third-party monitoring against the average concentration of the three indicators at each site.

As shown in Figure 4, comparing the slopes on the left and right sides of the above figures, after third-party monitoring the slopes of the curves for ammonia nitrogen ($NH_3$-N) and dissolved oxygen (DO) indicators decrease. However, the slope of the curve for permanganate ($COD_{Mn}$) index increased after third-party monitoring. If the absolute value of the slope of the curve decreases, it means that the change range of the absolute value of the t-statistic of the McCrary test decreases. The smaller the t-statistic, the more the null hypothesis $H_0$ is credible, and there is no significant difference between the left and right limits of the cut-off, and there is less possibility of data manipulation.

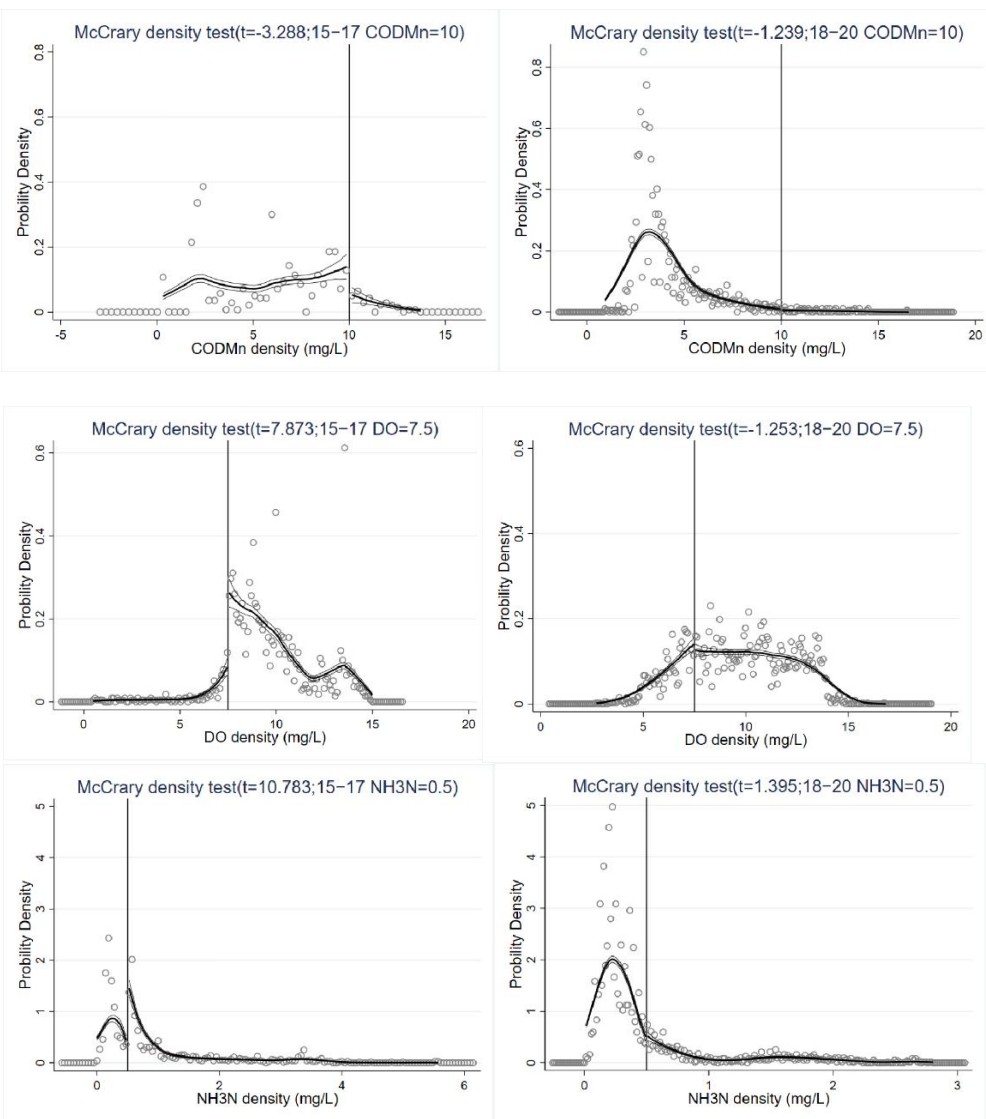

**Figure 3.** McCrary Density Test of different indicators.

The improvement of data quality after third-party monitoring mainly occurred in the two water pollution indices, ammonia nitrogen ($NH_3$-N) and dissolved oxygen (DO). According to the bidding documents of the National Surface Water Environmental Monitoring Network for manual monitoring of cross-section monitoring technical services issued by the China Environmental Monitoring Center, on-site monitoring by the third party may include water temperature, pH, dissolved oxygen (DO), and conductivity measurement. On-site monitoring data are uploaded to the environmental monitoring station on the same day, and the station is notified immediately if any abnormal data are found. Dissolved oxygen (DO) is included in the on-site monitoring project, and is more important than the other two indicators. In conclusion, third-party monitoring can reduce data manipulation and improve the accuracy of water environment data.

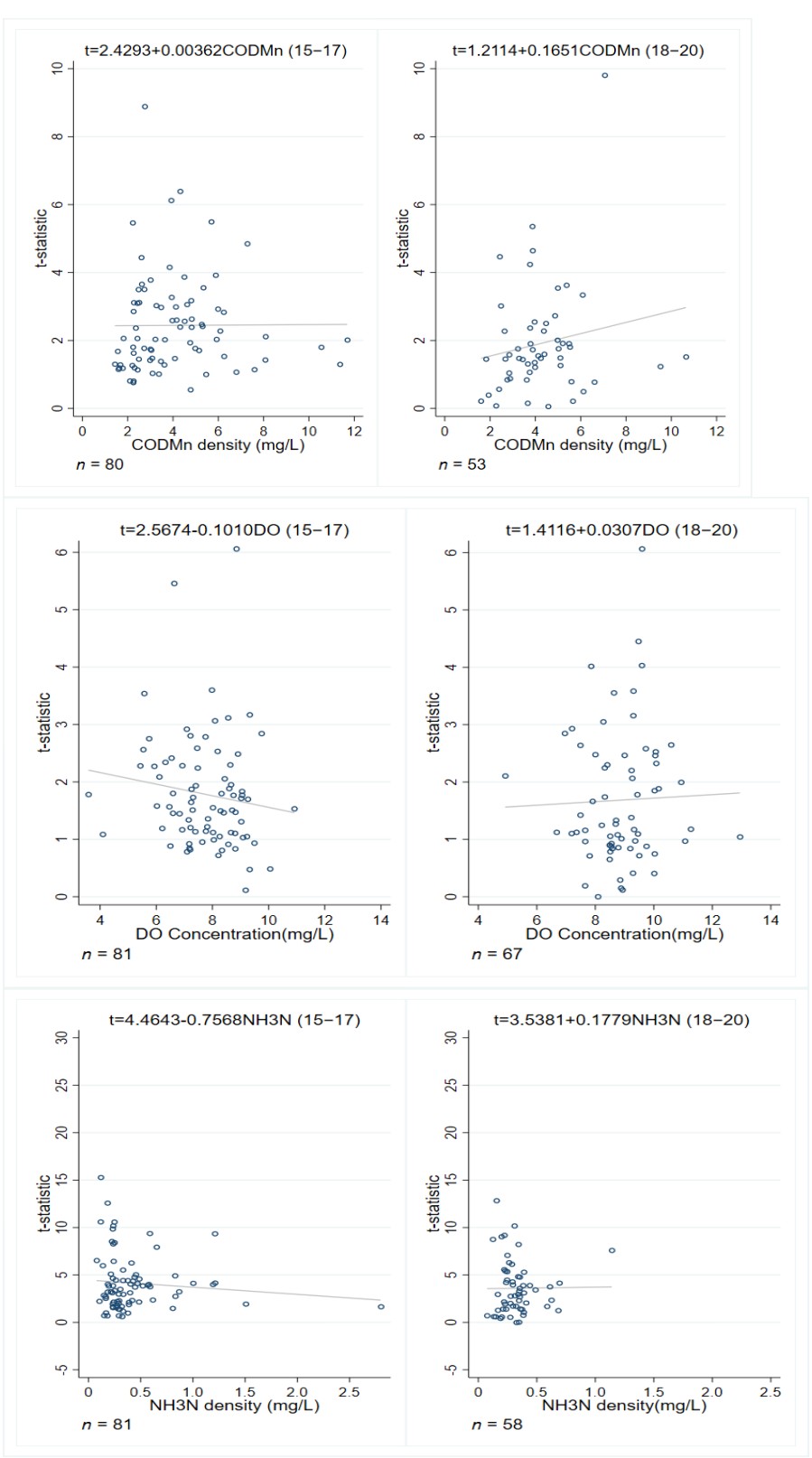

**Figure 4.** t-Statistics of the McCrary test of different indicators against the average pollution concentration.

*4.2. Robustness Test*

In the previous analysis of the results, water quality data from 2015 to 2020 released by surface water monitoring stations were more or less likely to be manipulated at each grading point, and the number of stations involved in data manipulation decreased significantly

after third-party monitoring. To illustrate the reliability of these empirical conclusions, robustness tests were carried out.

First, in the above empirical process, we used the regression discontinuity test of bin size and bandwidth calculated by default in the Stata program. Because the choice of bandwidth and bin size affects the test results to a certain extent [31], to ensure the robustness of the results, the bin size and bandwidth were manually changed and tested again. McCrary (2008) recommends a ratio of bandwidth to bin size a = h/b greater than 10. Therefore, we choose the dissolved oxygen (DO) and permanganate ($COD_{Mn}$) b to have a value of 0.1, h a value of 2, the ammonia nitrogen ($NH_3$-N) test b a value of 0.01, and h a value of 0.3, then the McCrary (2008) test was performed again. Compared with the unadjusted test results, the test results after adjusting the bin size and bandwidth had smaller changes in the t-statistic for each site, indicating that the results were reliable. Detailed results are shown in Table A2 in the Appendix A.

The cut-off selected in the above test was the graded point of the three indicators. In order to prove that the cut-off did not exist randomly, but at the graded point, we chose the value of the nongraded point. Since only dissolved oxygen was included in the on-site monitoring items of the three indicators, the dissolved oxygen index (DO) was tested when values of 1.5, 2.5, 4, 5.5, and 7 were the cut-off values. Scatter diagrams, as in Figure 5, show that the regression slopes at the nongraded points of each indicator before and after the third-party monitoring are small, and the change not significant. The result of the McCrary density test shows that the occurrence of data discontinuity in the above test was not random, but related to the standard limit of each grading point of the three indicators.

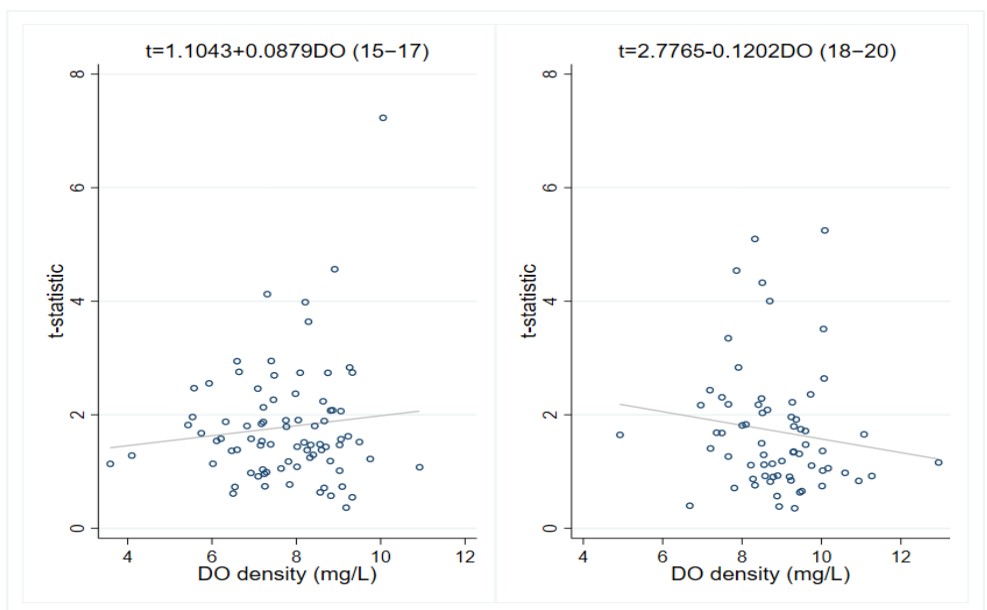

**Figure 5.** t-Statistics of the McCrary test of DO against the average pollution concentration.

Data from sites that did not participate in third-party monitoring should not be affected. Figure 6 shows the continuous changes in the dissolved oxygen index (DO) data of these sites. It can be seen that after third-party monitoring the slope increased, indicating that these sites were not affected by third-party monitoring. Our results illustrate the validity of our analyses and the robustness of the regression model in this paper.

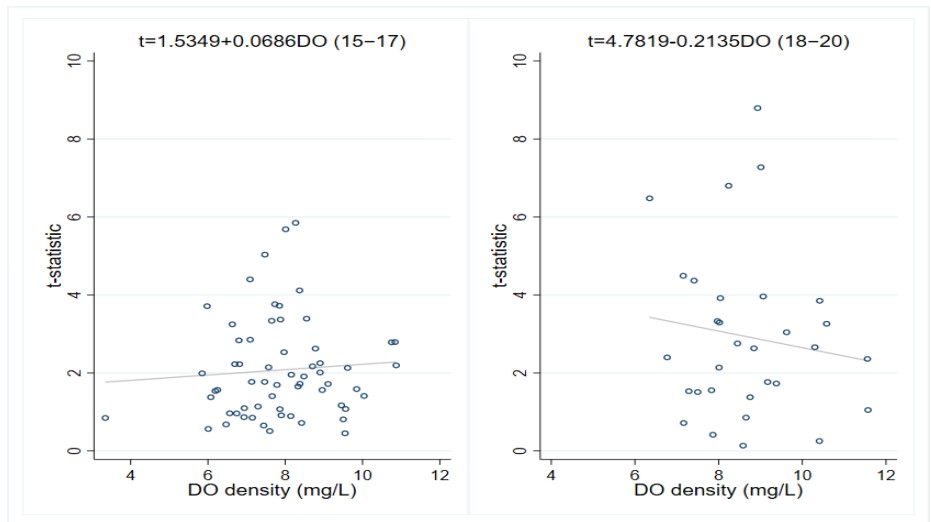

**Figure 6.** t-Statistics of the McCrary test of DO without third-party monitoring against the average pollution concentration.

## 5. Discussion

Environmental monitoring is the basic work of environmental protection, and the quality of environmental data seriously affects the process of government decision-making and policymaking. Inaccurate environmental data leads to inaccurate data analysis results and even affects the credibility of the government. The Ministry of Ecology and Environment has issued "The Three-year Action Plan for Quality Supervision and Inspection of Ecological and Environmental Monitoring (2018–2020)", stressing that a sound responsibility system for ensuring the quality of ecological and environmental monitoring data will be basically achieved by 2020, and that the problem of falsification of monitoring data will be effectively curbed.

The central government is the representative of the public interest of the entire country and society, and its policy-making goal is to maximize the interests of all people. According to the analysis of the environmental supervision organization system in Section 2, the central government entrusts surface water pollution control tasks to local governments, forming a principal-agent relationship between the central government and local governments. As far as the issue of environmental protection is concerned, the central government's main goal is to improve the environment, but local governments may consider how to obtain incentives from their superiors to pursue job promotions. Their ultimate goals are not the same, and the local government has the actual information at the regional level, and it is difficult for the central government to fully grasp the actions and information of the local government. Due to information asymmetry between the principal and the agent, the differences in the goals of the participating subjects and the conflicts between multiple project tasks, a principal-agent crisis often occurs. Third-party supervision refers to the participation of a third-party other than the principal and agent to supervise and manage the agent's behavior. Many scholars have found through empirical data testing that local data, especially environmental data, may be manipulated in the context of project system and performance evaluation. Niu et al. [14] focused on air quality monitoring, suggesting that involving third parties in environmental governance can provide independent external supervision, and reduce data manipulation by local governments. The research object of the current study was water quality data, and the effectiveness of third-party supervision in environmental monitoring was preliminarily proven by the McCrary density test. This mechanism of cooperation between government departments and social organizations is conducive to improving the efficiency of public service supply and the quality of public services.

In the past, the local monitoring model was monitored by the local environmental protection department and reported to the Central Environmental Monitoring Station. The central government evaluated the local government based on the data reported by the local government. This method of "who will be evaluated and who will monitor" is prone to administrative intervention, and there have been occurrences of concealing or falsifying data, which make it difficult to meet the current needs of environmental protection development. Withdrawal of the monitoring power of local governments with respect to the national surface water environment is a major decision towards the developing trend of environmental protection, and to deepen reform of the monitoring system. The central government has gradually taken over the power of monitoring surface water, breaking the local government's monopoly on monitoring data. The use of third-party organizations to collect water quality samples will only give full play to the role of social capital and save government management costs, but also ensure objective and fair monitoring of data to the greatest extent. At the same time, this also reflects that the country is opening up the environmental monitoring market, adding third-party monitoring companies to alleviate the current situation in which the monitoring capabilities of environmental protection departments cannot meet the needs of society and the government. The emergence of third-party monitoring agencies not only relieves the pressure on the government and related institutions, but also finds environmental quality problems faster, which is conducive to timely governance. The monitoring data is disclosed to the public in a comprehensive and timely manner, fully guaranteeing the people's right to know, participate, and supervise environmental data, and also provides basic support for water environmental protection work in various places.

It is inaccurate to evaluate water quality based only on physicochemical indicators. Different from Europe and the United States, the main water quality indicators used by the Chinese central government when assessing the surface water environment of local governments are physicochemical indicators. Judging from currently published data, third-party monitoring has improved the quality of water environment data and significantly improved the dissolved oxygen (DO) and ammonia nitrogen ($NH_3$-N) levels. In some formal assessment documents, the important assessment status of dissolved oxygen (DO) and ammonia nitrogen ($NH_3$-N) is clearly pointed out. According to the incentive theory, under the current technical limit, although the influence of third-party monitoring may be very limited, it is more effective than the original territorial monitoring method. It is recommended that the Chinese government refer to international standards, improve the water quality evaluation system, involve more water quality evaluation indicators and include other important indicators in on-site monitoring to achieve full control of river water quality. We hope that these proposals can arouse the government's attention. In future research, we will collect more information to improve shortcomings in this area.

## 6. Conclusions

This article is mainly based on the time-point water pollution data of 81 key automatic water quality monitoring points across the country from 2015 to 2020. The McCrary density test was used to detect whether public water pollution data underwent data manipulation at each level of each indicator, and compared the discontinuous changes of water environment data before and after third-party monitoring. The main research conclusions are the following. (1) The McCrary density test showed that 81 monitoring sites had more or less different degrees of data discontinuity from 2015 to 2020. The results revealed two modes of data manipulation: one to underreport the level of water pollution to make the water quality better, and the other to over-report the level to make the water quality worse. Data manipulation occurred at a relatively high proportion of the three classification points of class I and class II classification points. (2) By comparing the t-statistic of the McCrary test and the scatter plots drawn at each grading of each index from 2015 to 2020, it was found that the absolute slopes of the curve fitting index of dissolved oxygen (DO) and ammonia nitrogen ($NH_3$-N) decreased after third-party

monitoring. In addition, a decline in the variation range of the t-statistic indicated that third-party monitoring reduced local data manipulation behavior, improved the quality of water pollution data, and improved the accuracy of data. (3) The improvement of data quality after third-party monitoring mainly occurred in the two water pollution indicators of dissolved oxygen (DO) and ammonia nitrogen ($NH_3$-N), which indicated that third-party monitoring had a greater impact on the indicators valued in the assessment. More water quality monitoring indices should be included in on-site monitoring projects to achieve a comprehensive grasp of river water quality.

Our research provides a reference for third-party monitoring in environmental governance and proves that the participation of third-party organizations can have a deterrent effect on local governments, reducing local data manipulation behavior, and improving the accuracy of environmental data. The process of third-party monitoring involves the central monitoring station making plans, and the government making public bids. The third-party monitoring company that wins the bid is responsible for sample collection. This changes the original territorial monitoring model and transfers sample collection and data analysis to different units to avoid administrative intervention, cuts off the connection between local governments and self-reported data, increases the cost of local government data manipulation, and forces local governments to reduce data manipulation. The policy of third-party monitoring also reflects the transformation of China's environmental management system from a single, nonparticipatory model to a mixed and participatory model. The effectiveness of third-party monitoring in water environment monitoring indicates that, in the future, more third-party organizations can be involved in the formulation and implementation of environmental policies, and third-party organizations are encouraged to participate in the construction of public projects. The government can decentralize power appropriately, no longer needing to supervise and control the whole process of project construction, let more professional third-party organizations participate in the corresponding projects, and improve the government's administrative efficiency.

This article is a tentative study on the impact of third-party monitoring on data accuracy. The data in the empirical part of this article are based on the time-point water pollution data of 81 key automatic water quality monitoring points across the country from 2015 to 2020, with a time span of 6 years. The third-party monitoring of the water environment was officially implemented in October 2017, but the policy impact is a long-term process, and there is also a multistage game process between the central government and local governments. The research in this article is only preliminary proof that third-party monitoring can reduce data manipulation and improve the accuracy of water pollution data. It is clear that the current water quality indicators are only partial indicators and are still not perfect; certain effects have been seen. With the improvement of technology, China's water environment management has shifted from quantity management to quality management, and the central government is also preparing to take the indicators of hydromorphological and biological indicators into consideration. In the next step of the study, with the availability of data, we hope to supplement relevant data to better reflect changes in water quality. Quasi-experimental models can be added to make the research results more reliable.

There has been more research on air quality in China, and less research on water quality. Moreover, in the field of the water environment, the time for large-scale implementation of third-party monitoring is relatively short, and development is not perfect. There are relatively few articles on how to improve environmental quality and how to change China's environmental supervision system. This article provides some evidence for the efficiency of third-party monitoring in the water environment, and is a tentative study on the governance effect of policy innovation. On the one hand, it enriches empirical research in the field of the water environment. On the other hand, at the theoretical level, previous research mainly focused on the incompatibility of incentives between the central government and local governments, exposing the issue of data manipulation in the field of environmental supervision. There are few studies on the environmental

quality effects of major policy innovations. We have made an attempt to elucidate the effectiveness of policy innovation in water environment governance, hoping to provide the government and academia with useful information. We hope that there will be more policy and practice innovations in the future to promote the participation of the government, various organizations, social groups, and the public in environmental supervision to truly and comprehensively improve the quality of the water environment. On the other hand, we think that the cost of supervision must also be considered. More and comprehensive water quality indicators in the environmental monitoring process may mean higher policy implementation costs, and higher costs may directly hinder this, which will affect the implementation of the policy. The verification of these conjectures may require further exploration and research based on richer data and policy cases. We hope to have more in-depth thinking and discussion on this aspect in the future.

**Author Contributions:** Conceptualization, X.W.; formal analysis, M.H. and X.W.; funding acquisition, X.W.; methodology, M.H. and X.W.; supervision, X.W.; writing—original draft, L.Z., M.H. and X.W.; writing—review & editing, L.Z. and X.W. All authors have read and agreed to the published version of the manuscript.

**Funding:** This research was funded by the Research Project from Publicity Department of Zhejiang Province, grant number 1050QBN0121002G, and the National Natural Science Foundation of China, grant number 71803175; 71773114.

**Institutional Review Board Statement:** Not applicable.

**Informed Consent Statement:** Not applicable.

**Data Availability Statement:** The data in this study are available from the corresponding authors upon request.

**Acknowledgments:** The researchers kindly thank the Qingyue Open Environmental Data Center (https://data.epmap.org, accessed on 8 July 2020) for support on Environmental data processing.

**Conflicts of Interest:** The authors declare no conflict of interest.

## Appendix A

**Table A1.** t-Statistics for the McCrary test.

| Number | COD$_{Mn}$(15–17) I | II | III | IV | V | COD$_{Mn}$(18–20) I | II | III | IV | V |
|---|---|---|---|---|---|---|---|---|---|---|
| 1 | 6.564 | 0.94 | 2.314 | | | 6.288 | 3.699 | 2.743 | | |
| 2 | 1.705 | 2.536 | −0.887 | | | | 5.146 | 0.997 | 0.227 | 1.303 |
| 3 | −0.23 | 3.635 | −1.547 | 2.056 | 3.117 | | 0.791 | | | |
| 4 | −1.065 | | −1.544 | −3.3 | 1.306 | −0.79 | 0.06 | 3.346 | −0.668 | −1.317 |
| 5 | | −2.069 | 2.55 | −0.563 | 2.889 | | | | | |
| 6 | −6.318 | | 0.691 | | | −2.37 | | 0.539 | | |
| 7 | −2.144 | 2.038 | 1.063 | 0.315 | | 10.953 | −1.669 | 3.457 | | |
| 8 | −1.789 | 1.142 | 1.492 | | | −0.335 | 8.122 | −0.266 | −0.979 | −0.362 |
| 10 | | | | | | 6.825 | | 1.85 | 0.391 | |
| 11 | 3.286 | 1.808 | −6.056 | 1.561 | | −0.554 | 3.051 | | −0.138 | 0.527 |
| 12 | | 3.222 | 0.992 | | 1.103 | 1.778 | −1.235 | −2.271 | | |
| 13 | 5.685 | 1.727 | 0.362 | | | | 1.096 | 2.607 | −0.817 | −0.338 |
| 14 | | 1.117 | −0.781 | 1.215 | | −1.642 | 0.493 | −1.926 | | |
| 15 | −11.907 | −9.633 | −5.132 | | | | −2.704 | | 0.186 | |
| 16 | −0.577 | −0.358 | −0.724 | | | 4.521 | −1.394 | 2.279 | | |
| 17 | | −0.427 | 1.578 | | | 5.725 | 1.154 | −0.479 | −0.272 | |
| 18 | −2.674 | 6.234 | 2.51 | −1.422 | 1.336 | | 3.776 | 0.796 | 1.221 | −0.168 |
| 19 | 1.424 | 0.269 | 0.162 | −0.789 | −3.082 | 0.967 | | | 0.305 | 3.291 |
| 20 | 2.117 | 3.377 | −1.709 | | | 0.36 | 2.266 | | | |
| 21 | | −5.687 | 0.804 | 1.215 | | −4.621 | −1.919 | −4.425 | 1.185 | 0.381 |
| 22 | | 5.066 | | | −0.203 | 1.401 | 2.211 | 3.676 | | −0.347 |

**Table A1.** *Cont.*

| | | | | | | | | | | | |
|---|---|---|---|---|---|---|---|---|---|---|---|
| 23 | | 5.81 | −3.254 | | 0.121 | −1.234 | −3.969 | 2.445 | | |
| 24 | 0.494 | −0.879 | | −1.674 | | −0.67 | 2.757 | −1.859 | 1.111 | |
| 25 | 12.333 | −8.845 | −1.192 | 2.135 | | 11.057 | 7.484 | 3.105 | −1.507 | −0.081 |
| 26 | | 1.896 | 0.382 | | | 3.987 | −2.805 | −0.693 | 2 | |
| 27 | 2.374 | 2.144 | 3.297 | | | | 0.796 | 2.165 | | |
| 28 | | 3.69 | −1.013 | −1.11 | | −1.903 | 4.806 | 2.158 | −0.262 | |
| 29 | 5.292 | −2.412 | 2.46 | 1.551 | | | −1.491 | 0.552 | 0.287 | |
| 30 | −2.247 | −6.342 | 0.347 | | | | 0.686 | 3.562 | 0.951 | |
| 31 | 1.228 | 5.094 | 3.419 | 1.718 | 0.64 | −1.125 | −4.13 | −0.997 | 0.984 | |
| 32 | | −0.103 | −3.181 | 1.006 | | | 8.504 | 2.004 | | −0.134 |
| 35 | −0.454 | 2.456 | | | | | | | −0.866 | −0.826 |
| 36 | 0.769 | 1.632 | | | | | | −0.08 | | |
| 37 | 3.598 | −4.12 | −0.736 | 0.014 | −0.125 | | 1.156 | 1.076 | 1.84 | −0.114 |
| 38 | 1.808 | 0.202 | −0.824 | 2.85 | | | | | | |
| 39 | −0.839 | 4.487 | 2.377 | −1.226 | 1.249 | | −0.896 | −0.155 | 1.821 | 0.5 |
| 41 | −0.463 | 3.086 | | | | | 3.269 | | 0.743 | 1.266 |
| 42 | 1.03 | −1.343 | | | | | −5.953 | | 2.987 | |
| 43 | 6.58 | −4.354 | | | | | 2.585 | −0.581 | | |
| 46 | 1.838 | −4.012 | | −3.27 | 0.361 | | 9.546 | 0.717 | 2.334 | 0.772 |
| 47 | 3.772 | | 4.715 | −3.288 | | | 1.41 | 1.991 | −1.256 | |
| 48 | 5.435 | 2.353 | 0.993 | −0.79 | | | 0.76 | 1.983 | 0.719 | −1.609 |
| 53 | | −9.082 | | 3.704 | | | | | | |
| 55 | 3.245 | 0.826 | 0.823 | | | | | | | |
| 56 | 5.715 | 1.602 | | | | | 4.184 | −0.375 | | |
| 57 | 3.81 | 1.267 | 1.114 | | | | | | | |
| 58 | 2.328 | −1.412 | 0.946 | 1.241 | | | | | | |
| 59 | | | | | | | | | | |
| 60 | −0.121 | 1.969 | 0.347 | | | | | | | |
| 61 | | 13.307 | 2.68 | | 0.505 | | 4.72 | −2.539 | | |
| 70 | 1.5 | 6.95 | 0.786 | 1.542 | −0.637 | | | | | |
| 73 | 3.952 | 1.798 | 3.343 | | | | | | | |
| 74 | 0.849 | 8.051 | 0.093 | −0.914 | | | 19.438 | | | 0.184 |
| 78 | | 4.372 | 2.741 | | | | | | | |
| 79 | | 1.652 | −1.369 | 1.749 | −0.431 | | | | | |
| 80 | 1.591 | 9.522 | −0.502 | | | | | | −0.06 | |
| 81 | −0.371 | 1.983 | 1.128 | 2.658 | | | | | 0.5 | |
| 83 | −1.702 | 1.909 | 0.493 | −0.181 | | | | | | |
| 85 | 1.216 | 2.158 | 0.484 | | | | | | −0.157 | |
| 86 | 5.873 | −6.075 | −0.526 | | | | | | | |
| 87 | −0.847 | 2.247 | 2.988 | | | | | | | |
| 88 | | 4.115 | −3.785 | −9.183 | 2.324 | | | | 0.406 | 0.028 |
| 92 | 6.122 | 3.65 | −0.758 | | | | | | | |
| 93 | −1.709 | −2.767 | 3.03 | 7.63 | | | | | | |
| 95 | −5.55 | −0.484 | −0.612 | −4.79 | | | | | | |
| 96 | −2.019 | 3.219 | 0.002 | | | | | | | 0.569 |
| 97 | 3.229 | 3.571 | 1.054 | 0.297 | | | | | 1.767 | −1.192 |
| 98 | 6.499 | −2.133 | −0.345 | | | | | | | |
| 99 | 2.52 | | 0.019 | | | | | | | |
| 100 | −2.004 | 0.606 | | | | | | | | |
| 101 | 2.73 | 3.471 | | | | | | | 0.883 | |
| 102 | 8.098 | 4.76 | 0.473 | | | | | | | |
| 103 | 0.773 | −0.845 | | | | | | | | |
| 104 | −2.065 | | | | | | | | | |
| 105 | −1.525 | −1.299 | 1.475 | 8.179 | | | | | | |
| 106 | 3.118 | | | | | | | | | |
| 107 | −2.528 | | 1.087 | | | | | | | |
| 108 | −0.523 | −0.625 | 1.145 | | | | | | | |
| 111 | 1.216 | 1.381 | −0.988 | | | | | | | |
| 112 | 0.646 | 1.88 | 0.94 | | | | | | −0.397 | |
| 113 | −1.973 | 0.603 | | | | | | | −0.219 | |
| 115 | 1.836 | 1.531 | | | | | | | 1.344 | −1.574 |

**Table A1.** *Cont.*

| Number | DO(15–17) | | | | | DO(18–20) | | | | |
|---|---|---|---|---|---|---|---|---|---|---|
| | I | II | III | IV | V | I | II | III | IV | V |
| 1 | 2.078 | −0.548 | −0.454 | −1.999 | −7.018 | 0.722 | −0.044 | 0.843 | 0.654 | 1.004 |
| 2 | −0.111 | 0.318 | 1.528 | 1.384 | −2.272 | 0.467 | 0.503 | 1.247 | 0.775 | −0.601 |
| 3 | | | 1.069 | | −2.47 | | | | | |
| 4 | −1.587 | 2.204 | 1.146 | 3.271 | −0.809 | | 0.562 | −0.377 | 0.122 | 3.583 |
| 5 | −0.448 | 1.597 | 0.49 | 2.443 | −0.703 | | | | | |
| 6 | 1.111 | | | 1.504 | 1.05 | | | −1.293 | 0.293 | 0.665 |
| 7 | 0.175 | −0.117 | −2.601 | 3.186 | −3.356 | | | | 0.128 | −0.688 |
| 8 | 0.239 | 6.698 | −1.656 | 0.769 | 2.009 | −1.04 | | −1.02 | −1.391 | 6.471 |
| 10 | 0.437 | 0.974 | 1.979 | 0.029 | −3.127 | | | −1.262 | 2.798 | −2.144 |
| 11 | 1.894 | 1.264 | 0.56 | 0.153 | 0.065 | | | −0.202 | 1.684 | 5.508 |
| 12 | −0.479 | −1.148 | −0.071 | −1.322 | −1.219 | −0.97 | | −1.735 | 2.697 | 0.132 |
| 13 | 0.526 | −2.683 | 1.566 | 4.666 | −2.28 | 0.084 | | 1.511 | −0.462 | 2.364 |
| 14 | 4.299 | −3.543 | 3.367 | −1.578 | −0.04 | −1.134 | | 3.712 | 0.064 | −0.086 |
| 15 | −0.844 | 0.826 | −1.297 | 6.138 | −6.234 | | | 1.708 | 5.295 | 3.763 |
| 16 | −2.227 | 7.077 | −7.156 | 10.323 | 0.518 | −2.611 | | 2.46 | 3.249 | −5.908 |
| 17 | 0.861 | 1.111 | −0.821 | 0.896 | −2.345 | 0.511 | | 3.615 | −4.611 | 1.832 |
| 18 | 1.079 | 0.934 | −0.441 | −3.532 | 6.69 | | | −0.494 | | 4.16 |
| 19 | 0.345 | −1.33 | 0.073 | 3.751 | −1.302 | | | 0.74 | 0.634 | 6.03 |
| 20 | 1.787 | 3.718 | 5.619 | 2.525 | 0.135 | 1.869 | | 2.735 | 1.905 | −4.892 |
| 21 | 3.366 | 1.551 | −0.614 | 1.122 | 4.76 | −0.851 | | −0.503 | −1.192 | 3.159 |
| 22 | 1.236 | −0.652 | −0.395 | −0.548 | 4.503 | | | 1.213 | 1.142 | −1.186 |
| 23 | 0.14 | −1.345 | −3.497 | 2.187 | 1.41 | 0.826 | | −1.865 | 1.171 | 0.317 |
| 24 | 1.201 | −0.295 | 6.305 | −1.944 | 1.685 | 1.14 | | 0.1 | 0.938 | 1.35 |
| 25 | −0.564 | | −0.387 | 0.494 | 1.905 | 0.251 | | 1.992 | 0.063 | −1.06 |
| 26 | | | 0.523 | 2.497 | −2.379 | | | | 1.15 | 0.536 |
| 27 | 1.871 | −2.422 | 1.933 | 0.228 | −0.251 | 4.06 | | 4.086 | −6.708 | 1.227 |
| 28 | −0.393 | 1.3 | −1.889 | −0.669 | −1.597 | −1.537 | | −0.054 | −1.608 | 0.523 |
| 29 | 1.517 | −0.196 | 0.233 | 2.464 | −0.275 | 0.708 | | 3.288 | −0.729 | 2.405 |
| 30 | 2.926 | 0.685 | 1.244 | 1.989 | −7.189 | −1.119 | | 0.64 | 4.913 | −2.536 |
| 31 | 0.857 | 0.087 | 0.8 | −0.075 | −0.566 | | | 0.4 | −2.042 | −0.474 |
| 32 | 0.072 | −1.908 | 0.569 | −1.709 | 1.707 | | | 0.419 | 1.484 | 0.674 |
| 35 | −0.153 | −0.51 | | | 7.873 | | | 3.914 | 2.4 | −1.438 |
| 36 | | −2.916 | 0.278 | 1.266 | −3.77 | | | −2.398 | 3.923 | −7.044 |
| 37 | | | −0.378 | −1.306 | 7.672 | | | | 0.247 | −2.101 |
| 38 | | | | 1.861 | 5.343 | | | | | |
| 39 | 0.25 | −0.616 | 2.361 | −3.248 | −1.376 | −1.473 | | 3.052 | 1.485 | 0.653 |
| 41 | | −0.878 | −1.166 | 0.193 | −1.973 | | | −0.549 | −0.08 | 2.62 |
| 42 | | | | | 0.487 | | | | −1.177 | −2.596 |
| 43 | 0.353 | −0.977 | | 2.894 | −3.125 | | | | | 2.65 |
| 46 | −1.266 | 1.129 | 0.5 | −0.346 | −0.813 | 3.087 | | −1.354 | −2.016 | 6.18 |
| 47 | | | 1.277 | 0.86 | 1.017 | 1.006 | | 0.759 | 0.884 | −0.879 |
| 48 | 1.74 | 2.027 | 4.774 | −0.506 | 0.334 | 1.026 | | −0.264 | 0.857 | 1.756 |
| 53 | 1.314 | −0.242 | 1.447 | −1.56 | 0.609 | | | | | |
| 55 | −0.28 | 2.713 | 0.85 | 3.898 | −0.176 | | | | | |
| 56 | | | | 3.772 | −1.41 | 1.499 | | 2 | 1.851 | 6.385 |
| 57 | | 1.204 | 0.96 | 3.102 | 4.688 | | | | | |
| 58 | −0.857 | −1.076 | 1.531 | −0.345 | −0.622 | | | | | |
| 59 | | | | | | | | | | |
| 60 | | | −0.193 | | 0.04 | | | | | |
| 61 | 0.261 | −0.298 | −0.244 | 2.176 | 2.63 | −0.382 | | | 1.625 | −1.038 |
| 70 | −0.897 | 3.579 | 5.051 | −0.54 | 0.386 | | | | | |
| 73 | −1.13 | 0.52 | −0.008 | −2.959 | −2.878 | | | | 0.295 | |
| 74 | −0.624 | | −3.688 | −0.586 | −1.241 | 1.024 | | −0.958 | 2.011 | −3.996 |
| 78 | 0.646 | 0.192 | −2.435 | 0.219 | 4.063 | | | | | |
| 79 | 2.267 | 1.409 | 0.283 | −1.071 | −0.412 | | | | | |
| 80 | | | | 1.455 | −0.534 | | | | | −0.153 |
| 81 | −0.287 | 0.197 | 0.964 | 1.496 | −4.626 | 2.534 | | −1.221 | | 2.858 |
| 83 | 3.679 | −1.153 | 2.15 | 0.151 | | −1.011 | | −2.802 | 3.469 | −1.152 |
| 85 | 1.871 | −1.176 | 9.598 | −0.296 | 4.773 | −1.082 | | 0.877 | 0.787 | 1.759 |
| 86 | 1.624 | −0.165 | 0.622 | −1.038 | 2.271 | | | | 0.268 | 3.436 |
| 87 | −0.157 | | 3.305 | −0.827 | 0.141 | | | | 2.787 | 5.282 |
| 88 | −0.7 | 1.578 | −0.477 | 1.51 | 0.512 | | | 1.662 | 0.25 | 3.314 |
| 92 | | | 2.532 | −1.894 | −0.519 | | | | | 0.003 |
| 93 | | −1.544 | −4.463 | 4.054 | 2.626 | | | | | |
| 95 | | | −2.125 | −0.39 | 1.909 | | | | | |

**Table A1.** *Cont.*

| Number | I | II | III | IV | V | I | II | III | IV | V |
|---|---|---|---|---|---|---|---|---|---|---|
| 96 | | −0.135 | 1.108 | 4.61 | −0.364 | 1.508 | | 4.661 | | −2.989 |
| 97 | | | 1.944 | 0.862 | −3.004 | −0.005 | | | | 2.535 |
| 98 | | 1.54 | −1.115 | 1.142 | −2.032 | 0.748 | | −0.817 | | 1.332 |
| 99 | | | | | 6.062 | | | | | 6.067 |
| 100 | | 0.136 | 0.713 | −1.941 | 0.873 | −0.358 | | 0.031 | | 0.851 |
| 101 | | | | −0.532 | 4.067 | | | 0.089 | | 2.106 |
| 102 | −0.761 | | 5.659 | −2.494 | 0.066 | | | | | −2.249 |
| 103 | | | | −0.254 | 3.653 | −0.691 | | 1.639 | | −1.674 |
| 104 | | | | | 0.723 | | | | | 0.121 |
| 105 | 0.549 | 1.26 | 8.4 | 1.196 | −3.204 | | | | | |
| 106 | | | 3.227 | −0.175 | | | | | | −2.53 |
| 107 | −0.93 | −0.978 | −2.951 | −0.295 | −8.79 | | | | | 1.058 |
| 108 | | | −0.973 | 2.33 | 2.011 | | | | | 0.789 |
| 111 | | | −0.331 | 3.206 | −1.661 | | | | | −0.715 |
| 112 | | | 1.424 | −0.055 | 1.295 | 1.173 | | | | 0.631 |
| 113 | | | 0.284 | 2.69 | −1.375 | | | | | −1.126 |
| 115 | | | 0.305 | 1.904 | −0.259 | | | | | 0.193 |

| Number | NH$_3$-N(15–17) I | II | III | IV | V | NH$_3$-N(18–20) I | II | III | IV | V |
|---|---|---|---|---|---|---|---|---|---|---|
| 1 | 16.134 | | | −0.705 | | 11.182 | 4.373 | 0.526 | | |
| 2 | 13.138 | −4.842 | 4.433 | −0.291 | −1.208 | 15.398 | | 1.211 | 0.402 | −0.125 |
| 3 | −0.113 | 3.115 | | 0.957 | | | | | | |
| 4 | 12.24 | 1.214 | 4.058 | 0.842 | | 9.349 | −6.116 | | −0.45 | |
| 5 | 10.462 | 4.162 | 2.287 | 0.535 | | 0 | | | | |
| 6 | 8.639 | | 0.686 | | | 12.82 | −7.279 | | | 1.142 |
| 7 | −4.948 | | | −0.497 | | 0.128 | | 0.744 | | |
| 8 | 3.612 | −5.176 | | | 3.716 | −2.522 | −1.178 | −1.552 | −1.481 | 1.735 |
| 10 | −0.991 | 1.67 | 0.084 | | 0.086 | 9.694 | 2.884 | | 1.247 | 2.067 |
| 11 | −5.888 | −8.21 | 1.886 | 0.732 | 3.958 | 10.342 | −2.026 | 0.661 | 0.058 | 1.027 |
| 12 | 11.327 | −7.441 | | | | 8.893 | −7.657 | 1.093 | 1.186 | 0.68 |
| 13 | −2.535 | 0.9 | | −0.823 | | 1.537 | 5.227 | | | −2.138 |
| 14 | 0.86 | 3.521 | 1.78 | −0.721 | 2.847 | −20.62 | | 2.162 | −1.528 | 0.937 |
| 15 | −15.62 | 4.106 | | | | −10.413 | | | 0.448 | |
| 16 | −5.341 | 6.339 | 2.827 | −1.719 | | 4.362 | 3.224 | 2.088 | 1.166 | −0.72 |
| 17 | −0.334 | 3.472 | 0.633 | 0 | 0.51 | −10.846 | −13.813 | | −2.466 | |
| 18 | −9.109 | 2.226 | 4.788 | 0.323 | | −34.35 | | | −2.04 | −2.141 |
| 19 | 0.01 | 0.514 | 2.042 | −0.074 | 0.453 | −5.595 | | 3.594 | −2.337 | 0.958 |
| 20 | 8.928 | | 0.522 | −0.656 | | 12.263 | −0.378 | | | 1.852 |
| 21 | 2.103 | −13.822 | 3.065 | 0.81 | 0.222 | 8.584 | 4.146 | 5.424 | | 0.986 |
| 22 | 5.823 | 1.551 | 1.613 | 1.585 | 1.295 | 3.907 | 4.024 | 2.402 | 1.684 | 1.543 |
| 23 | 16.66 | −11.551 | 0.929 | 0.835 | −1.407 | 3.698 | | −0.228 | 2.45 | −0.533 |
| 24 | −12.817 | | 3.374 | −0.727 | −0.798 | 4.636 | | | −3.805 | |
| 25 | −5.809 | −1.049 | | 1.453 | −0.551 | 7.724 | 3.885 | 2.765 | −1.206 | |
| 26 | 6.165 | | 0.856 | | | 6.142 | | −1.063 | | |
| 27 | 3.439 | −2.338 | −0.342 | | | 7.312 | 6.227 | −0.103 | 1.94 | 0.968 |
| 28 | 8.674 | −6.99 | 1.354 | 2.107 | −0.525 | 14.611 | | 0.588 | −3.227 | |
| 29 | −0.106 | 2.328 | 2.315 | −0.009 | | −8.804 | | | −2.714 | 1.902 |
| 30 | 2.224 | −3.433 | −0.125 | 0.99 | | −2.516 | 3.718 | −0.073 | −0.057 | −0.098 |
| 31 | 1.384 | −6.131 | | −0.652 | −0.474 | −15.483 | −11.006 | | −1.09 | |
| 32 | 7.076 | −4.957 | 1.506 | −1.189 | −1.509 | 6.794 | 1.838 | −1.557 | 0.607 | 0.972 |
| 35 | 16.381 | −4.811 | | | | 12.475 | −11.799 | | | 0.41 |
| 36 | 4.814 | 4.469 | | −0.142 | | −6.947 | | 1.007 | | 0.935 |
| 37 | 1.567 | | | | | −0.582 | | | | |
| 38 | 3.705 | −2.957 | −2.462 | | 0.193 | | | | | |
| 39 | 9.017 | 10.783 | 3.466 | 0.329 | −1.03 | 10.677 | 1.395 | 1.217 | 2.44 | 1.432 |
| 41 | 8.171 | | 0.765 | | | 22.33 | −8.157 | | | 0.059 |
| 42 | 3.535 | −0.934 | | | | −23.302 | | −2.434 | 0.55 | |
| 43 | −19.937 | | 1.282 | | | −18.956 | | −1.408 | 1.169 | −0.78 |
| 46 | −3.249 | 4.765 | 4.041 | | | −3.225 | −12.258 | −0.43 | −2.05 | 2.81 |
| 47 | 3.734 | 0.736 | 2.723 | 0.369 | | −1.627 | 5.641 | | 0.527 | −0.45 |
| 48 | 16.687 | −15.162 | | −2.06 | −3.524 | −22.633 | −6.894 | | 0.645 | 0.228 |
| 53 | 13.178 | −5.164 | −1.013 | | | | | | | |
| 55 | 3.253 | | | | | | | | | |
| 56 | 15.17 | | −1.408 | | | | | | | |
| 57 | −1.763 | | | | | | | | | |
| 58 | 9.815 | 3.591 | −0.674 | 0.854 | | | | | | |

**Table A1.** *Cont.*

| Number | I | II | III | IV | V | I | II | III | IV | V |
|---|---|---|---|---|---|---|---|---|---|---|
| 59 | | | | | | | | | | |
| 60 | 2.807 | | | 0.511 | | | | | | |
| 61 | 3.009 | | | | −0.136 | | | | | |
| 70 | 8.837 | −4.838 | 1.385 | | 0.004 | | | | | |
| 73 | 12.12 | −18.162 | 4.116 | −1.857 | 3.457 | | | | | |
| 74 | 12.689 | −7.835 | −3.293 | 1.017 | −0.363 | | | −0.031 | | |
| 78 | 10.126 | | 0.939 | | | | | | | |
| 79 | −2.202 | 2.038 | 0.184 | 0.638 | 3.226 | | | | | |
| 80 | 3.926 | 2.681 | | 0.297 | | | | 0.022 | −1.077 | |
| 81 | −4.611 | | | | 1.331 | 2.442 | 0.069 | −1.774 | | |
| 83 | −3.439 | 0.608 | 1.334 | −1.111 | 0.944 | 1.815 | −0.805 | 1.65 | −0.772 | |
| 85 | 4.109 | 2.036 | 4.573 | 2.545 | 0.529 | 0.905 | 0.625 | | | |
| 86 | −0.467 | | 1.07 | 1.052 | −0.25 | | 2.018 | | | |
| 87 | 16.116 | | −0.969 | | | | −1.419 | | | |
| 88 | 7.146 | 3.271 | 1.361 | 0.196 | | 3.057 | 1.262 | −0.029 | 1.173 | |
| 92 | 14.107 | | 0.026 | −0.816 | 0.468 | | | | | |
| 93 | 5.373 | 3.763 | | | −0.274 | | | | | |
| 95 | 7.952 | | 0.132 | | | | | | | |
| 96 | 11.113 | −1.41 | 0.732 | | | −4.037 | | 0.887 | −0.213 | |
| 97 | −5.592 | | | 0.135 | | | | −0.601 | | |
| 98 | 3.052 | 2.906 | | −1.297 | 1.253 | −6.496 | 3.118 | | 1.672 | |
| 99 | 6.538 | | | | | −0.718 | | | | |
| 100 | −11.757 | | | | −0.227 | | | | | |
| 101 | 4.081 | −4.653 | −0.933 | | | | | | | |
| 102 | 10.147 | | | 0.283 | 1.229 | | 2.175 | | | |
| 103 | 12.591 | | | | | | | −1.947 | | |
| 104 | −15.288 | | | | | | | | | |
| 105 | 5.475 | −7.227 | | −0.596 | −2.132 | | | | | |
| 106 | 3.317 | 1.849 | | −0.701 | | −1.073 | | | | |
| 107 | 8.934 | 1.279 | | | | −4.452 | | | 1.089 | |
| 108 | −4.344 | | | −0.748 | | 2.335 | 0.478 | | | |
| 111 | 5.055 | | −2.396 | −0.201 | −0.984 | | | | | |
| 112 | 0.157 | | 1.39 | −0.651 | | | | | | |
| 113 | −1.712 | | | 1.074 | 0.265 | 0.617 | | | | |
| 115 | 10.187 | | | | | | | | | |

**Table A2.** t-Statistics for the McCrary test after changing the bin size and bandwidth.

| Number | COD$_{Mn}$(15–17) | | | | | COD$_{Mn}$(18–20) | | | | |
|---|---|---|---|---|---|---|---|---|---|---|
| | I | II | III | IV | V | I | II | III | IV | V |
| 1 | 1.931 | −8.499 | 5.848 | 0.855 | | 10.513 | −3.717 | | | |
| 2 | 0.215 | 5.784 | −5.43 | | | | 5.195 | −1.186 | 0.006 | 1.395 |
| 3 | −0.227 | 2.747 | −0.377 | 3.335 | 2.657 | | 0.821 | | | |
| 4 | −1.449 | | −2.399 | −2.934 | 1.291 | −0.471 | 0.659 | 3.158 | −0.608 | −1.852 |
| 5 | | −of1.534 | 2.026 | −0.721 | 0.774 | | | | −0.713 | |
| 6 | −3.33 | 3.781 | 0.907 | | | −8.054 | | −1.046 | | |
| 7 | −2.801 | 2.965 | −1.16 | | | 8.578 | −1.493 | 3.514 | −0.352 | |
| 8 | −1.689 | 0.942 | 1.144 | | | −0.806 | 7.113 | −1.616 | −1.064 | −0.367 |
| 10 | | | | | | 9.601 | | −0.197 | 0.327 | |
| 11 | 3.081 | 2.172 | −8.512 | 1.617 | | −1.597 | 1.484 | | −0.115 | 0.292 |
| 12 | | 2.735 | 1.73 | −0.33 | 1.062 | 1.564 | −1.472 | −1.917 | | |
| 13 | −1.721 | −1.073 | | | | | 0.677 | 3.633 | −0.539 | |
| 14 | | 1.426 | −0.791 | 1.794 | | −0.467 | −2.734 | −2.925 | | |
| 15 | −11.324 | −9.287 | −5.774 | | | −5.826 | −14.06 | | | |
| 16 | | 0.002 | −5.286 | | | −0.163 | −1.104 | −4.121 | | |
| 17 | | −0.442 | −0.571 | 1.236 | | 5.05 | 0.574 | −1.13 | −0.215 | |
| 18 | −1.765 | 4.9 | 4.194 | −0.71 | 0.465 | 2.1 | 4.001 | −1.098 | 1.117 | −0.185 |
| 19 | 2.41 | −0.244 | −1.62 | −0.992 | −2.471 | 1.086 | | | 0.094 | 1.593 |
| 20 | 4.661 | 3.74 | −2.91 | | | −0.696 | −2.072 | | | 0.273 |
| 21 | | −5.677 | −1.802 | | | −4.57 | −1.757 | −11.112 | 1.157 | 0.228 |
| 22 | | 9.075 | −6.406 | −0.345 | | 2.203 | 0.638 | 1.74 | | −0.309 |

**Table A2.** *Cont.*

|  |  |  |  |  |  |  |  |  |  |  |
|---|---|---|---|---|---|---|---|---|---|---|
| 23 | 0.872 | 4.241 | −2.515 |  | 0.116 | −1.197 | −4.75 | 2.131 |  |  |
| 24 | −0.376 | −2.845 |  | −1.722 |  | −1.903 | 2.359 | −2.329 | 1.075 |  |
| 25 |  | −8.26 | −0.935 | 1.883 |  | 8.669 | 4.887 | 3.125 | −1.493 | −0.146 |
| 26 |  | −4.316 |  |  |  | 2.687 | −3.027 | −1.854 | 1.952 |  |
| 27 | 5.354 | 2.405 | 3.423 |  |  |  | −0.39 | −2.034 |  |  |
| 28 |  | 4.498 | −4.214 | −1.283 |  | −4.57 | 5.391 | 1.975 |  |  |
| 29 | 5.376 | −1.598 | 1.999 | 1.712 |  |  | −2.916 | 0.429 | 0.937 | −0.854 |
| 30 | −4.943 | −11.476 | 1.214 |  |  |  | −5.591 |  | 1.37 |  |
| 31 | 0.723 | 4.356 | 3.323 | 1.68 | 0.596 |  | 1.038 | −6.511 |  |  |
| 32 |  |  | 0.286 | 4.198 | 1.083 |  | 12.449 | 0.08 |  | −0.381 |
| 35 | 1.436 | 1.229 |  |  |  | 19.069 |  | 2.277 | −1.346 | −0.615 |
| 36 | 1.034 |  |  |  |  | 1.903 |  | −0.738 |  |  |
| 37 | 3.833 | −5.719 | −1.035 | −0.152 | −0.081 | −1.615 | 1.789 | 0.083 | 1.715 | −0.36 |
| 38 | 2.383 | −1.114 | −0.735 | 2.791 |  |  |  |  |  |  |
| 39 | −0.195 | 4.008 | 2.062 | −1.217 | 1.302 | 3.635 | −0.978 | −0.601 | 1.819 | 0.855 |
| 41 | 2.962 | 0.266 |  |  |  | 4.331 | −2.872 |  | 0.693 | 1.228 |
| 42 | −6.855 |  |  |  |  | −2.117 | −6.246 |  | 3.173 |  |
| 43 | 11.667 |  |  | 0.764 |  | 16.915 | −15.62 |  |  |  |
| 46 |  | −2.616 |  | −2.608 | 0.276 | −4.822 | 10.369 | 0.167 | 2.208 | 0.548 |
| 47 | 4.76 |  | 5.02 | −2.114 |  | 7.537 | −2.998 | 1.836 | −1.298 |  |
| 48 | 5.415 | 2.282 | 0.922 | −1.33 |  | 0.539 | 0.788 | 2.432 | 0.696 | −1.727 |
| 53 |  | −10.577 |  | 3.939 |  |  |  |  |  |  |
| 55 | 1.449 | −1.144 |  |  |  |  | −1.292 |  |  |  |
| 56 | 19.183 |  |  |  |  | 19.103 |  | −0.793 |  |  |
| 57 | 4.006 | 6.824 |  |  |  | 0.793 | 0.402 |  |  |  |
| 58 | 1.057 | −2.947 | −0.576 | 3.834 |  |  |  |  |  |  |
| 59 |  |  |  |  |  |  |  |  |  |  |
| 60 | −4.901 | 1.413 | 0.098 |  |  |  |  |  |  |  |
| 61 |  | 10.628 | 0.39 | 2.441 | 0.552 |  |  | 8.916 | −2.228 |  |
| 70 | −1.278 | 7.242 | 0.429 | 1.108 | −0.532 |  |  |  |  |  |
| 73 | 2.688 | −0.386 | 3.164 |  |  |  | −0.081 |  |  |  |
| 74 | −0.202 | 9.805 | −1.374 | −0.868 |  |  | 20.954 | 2.875 | 0.766 | 0.191 |
| 78 |  | 7.158 | 1.552 |  |  |  |  |  |  |  |
| 79 |  | 1.001 | −1.165 | 2.148 | −0.315 |  |  |  |  |  |
| 80 |  | 5.853 | 3.051 |  |  |  | 4.199 |  | 0.016 |  |
| 81 | −1.069 | 2.183 | 0.924 | 2.221 |  | 1.695 | 0.498 | −1.853 | −0.784 |  |
| 83 | −1.5 | 1.78 | 0.322 | 1.272 | 1.485 |  | 6.718 | −1.211 |  |  |
| 85 |  | −2.472 |  |  |  | 12.55 | −6.671 | 2.438 | −0.593 |  |
| 86 | 6.915 | −6.844 | 0.593 |  |  | 11.183 | −0.211 | 1.245 |  |  |
| 87 | 5.61 | −3.901 | 6.462 |  |  | 19.079 | −5.022 |  |  |  |
| 88 |  | 2.171 | −0.002 | −7.008 | 1.851 | 3.679 | 6.157 | −1.686 | 0.305 | 0.052 |
| 92 | 15.777 | 5.645 | −0.597 |  |  | 13.147 |  |  |  |  |
| 93 | −1.679 | −1.97 | 1.184 | 6.645 |  |  |  |  |  |  |
| 95 | −7.711 | −2.229 | −1.076 | −5.032 |  | −1.165 |  |  |  |  |
| 96 | 6.515 | −0.134 |  |  |  | 5.675 | 1.304 |  |  | 0.645 |
| 97 | 12.912 | 1.764 |  | 0.582 |  | −4.152 | 0.457 | −0.906 | 1.536 | −1.178 |
| 98 | 4.721 | −3.355 | −4.724 |  |  | 9.848 | −11.816 |  |  |  |
| 99 | 3.458 |  | −0.413 |  |  | 12.631 |  |  |  |  |
| 100 | −12.199 | −0.05 |  |  |  | −17.441 | 2.87 | 1.118 |  |  |
| 101 | 2.552 | 5.143 |  |  |  | 17.52 | −1.338 |  | 0.951 |  |
| 102 | 8.66 | 4.07 | 1.313 |  |  | −21.866 | −1.702 |  |  |  |
| 103 | −3.221 |  | 0.564 |  |  | −14.001 | 4.792 | 0.48 |  |  |
| 104 | −10.823 |  | −0.27 |  |  | −24.333 |  | 0.872 | 0.182 |  |
| 105 | −4.81 | −1.483 | 2.282 | 9.395 |  |  |  |  |  |  |
| 106 | 4.363 |  |  |  |  | 20.021 |  |  |  |  |
| 107 | −4.983 |  | 0.967 |  |  | 12.206 |  | 0.643 |  |  |
| 108 | −2.878 | 3.158 |  |  |  | −0.464 |  | 0.201 |  |  |
| 111 | −5.073 |  | −0.687 |  |  | 1.567 | 2.179 |  |  |  |
| 112 | −6.428 |  | 0.787 |  |  | −6.422 | 6.875 | 1.775 | −0.496 |  |
| 113 | −20.087 |  |  |  |  | −21.261 |  | −0.178 | −0.179 |  |
| 115 | −19.752 |  |  |  |  | 18.173 | 9.358 | 3.178 | 1.359 | −1.672 |

Table A2. *Cont.*

| Number | DO(15–17) | | | | | DO(18–20) | | | | |
|---|---|---|---|---|---|---|---|---|---|---|
| | I | II | III | IV | V | I | II | III | IV | V |
| 1 | 3.157 | −0.574 | −1.474 | 1.363 | −5.947 | 0.643 | −0.687 | 0.545 | 0.615 | 1.313 |
| 2 | 0.024 | 0.368 | 1.247 | 0.726 | −2.617 | 0.409 | 0.46 | 1.164 | 0.437 | −0.74 |
| 3 | | | | | −1.857 | | | | | |
| 4 | −0.65 | 2.974 | 1.532 | 3.417 | −0.631 | | 0.583 | −2.032 | 0.491 | 3.215 |
| 5 | −0.548 | 1.605 | 1.038 | 2.772 | 0.129 | | | | | |
| 6 | 1.01 | | | 5.08 | −0.599 | | 0.476 | −1.162 | 0.613 | 0.509 |
| 7 | | | | 4.389 | −4.059 | | | | 2 | −1.038 |
| 8 | −0.355 | 6.851 | −1.642 | 0.718 | 1.733 | −1.375 | −1.152 | −1.182 | −1.809 | 5.612 |
| 10 | 0.457 | 1.017 | 2.237 | −0.833 | −1.978 | | | | 5.392 | −5.869 |
| 11 | 1.644 | 0.893 | 0.162 | 0.134 | −0.021 | | | −0.306 | 1.651 | 5.396 |
| 12 | 0.186 | −1.859 | 0.085 | −1.234 | −1.542 | −0.43 | −0.689 | −2.146 | 3.386 | −0.007 |
| 13 | −0.835 | −0.737 | −1.308 | 5.856 | −4.683 | | 3.039 | 0.733 | −0.884 | 2.06 |
| 14 | 5.284 | −4.015 | 1.704 | −1.669 | 0.079 | −0.87 | 0.24 | 3.134 | 0.107 | −0.164 |
| 15 | | 0.428 | −2.841 | 5.6 | −5.004 | | | 1.802 | 5.007 | 1.191 |
| 16 | −1.412 | −0.677 | −6.148 | 10.086 | −0.232 | −2.089 | 1.94 | 1.248 | 2.604 | −5.135 |
| 17 | 0.745 | 1.247 | 0.158 | 0.847 | −2.651 | 2.384 | 1.334 | 4.429 | −5.699 | 1.107 |
| 18 | 1.71 | 0.178 | −0.179 | −3.187 | 6.041 | | 0.541 | −0.338 | | 4.759 |
| 19 | 0.367 | −1.269 | −1.246 | 4.088 | 0.715 | | −1.72 | 0.277 | 1.918 | 5.649 |
| 20 | 1.129 | 4.331 | 1.35 | 2.26 | 3.453 | 3.287 | 1.87 | 2.6 | 2.054 | −6.023 |
| 21 | 3.137 | 1.495 | −0.65 | 1.452 | 4.57 | | −0.474 | −2.211 | −1.004 | 2.802 |
| 22 | 1.181 | −0.408 | −0.489 | −0.806 | 3.17 | | | 0.998 | 0.258 | −1.54 |
| 23 | −0.562 | −1.146 | −3.528 | 2.239 | 1.278 | 1.035 | 0.226 | −1.912 | 1.167 | 0.241 |
| 24 | 1.401 | −0.397 | 6.259 | −1.373 | 0.314 | 0.92 | −0.222 | 0.072 | 0.892 | 0.542 |
| 25 | −0.671 | | −1.139 | 0.211 | 1.823 | 0.061 | | 1.583 | −0.543 | −1.573 |
| 26 | | | 0.691 | 3.467 | −4.722 | | 0.909 | | 1.503 | −1.011 |
| 27 | 1.683 | −2.439 | 1.749 | 0.831 | 0.13 | 3.886 | −1.302 | 3.74 | −6.842 | 0.667 |
| 28 | −0.429 | 1.09 | −1.4 | 0.902 | −2.06 | −1.855 | 1.003 | −0.003 | −1.733 | 0.22 |
| 29 | 1.784 | −0.061 | −0.48 | 3.109 | −0.181 | 0.36 | 1.803 | 3.31 | −0.213 | 2.098 |
| 30 | 2.819 | 0.6 | 0.154 | 2.671 | −6.875 | −1.445 | 1.022 | 0.604 | 3.795 | −2.697 |
| 31 | 0.887 | 0.069 | 0.127 | −0.034 | −0.593 | | | 0.722 | −2.141 | −0.514 |
| 32 | 0.307 | −1.66 | 0.344 | −1.145 | 1.47 | | −0.783 | 0.078 | 1.764 | 0.552 |
| 35 | −0.19 | −0.522 | −1.181 | | 8.809 | | 0.037 | 4.001 | 2.443 | −1.586 |
| 36 | | −2.739 | 0.093 | 1.436 | −3.873 | | 4.151 | −1.266 | 2.35 | −5.265 |
| 37 | | | −0.39 | | 13.734 | | | | −0.981 | 5.361 |
| 38 | | | | 3.459 | 7.602 | | | | | |
| 39 | 0.414 | −1.951 | 2.829 | −3.533 | −1.428 | −1.945 | −0.11 | 3.017 | 0.785 | 0.29 |
| 41 | | −1.534 | −0.455 | 0.288 | −1.982 | | 1.109 | −2.527 | 0.436 | 1.835 |
| 42 | | | | | 0.354 | −0.66 | | | 1.526 | −2.874 |
| 43 | 0.924 | −0.963 | | 5.655 | −4.099 | | | | | 3.843 |
| 46 | −0.767 | 0.963 | 0.464 | −0.358 | −0.544 | 2.032 | 0.404 | −1.138 | −0.145 | 1.355 |
| 47 | | −0.185 | 0.379 | 1.35 | 0.949 | 1.124 | −0.821 | 0.65 | 0.768 | −0.916 |
| 48 | 1.715 | 1.205 | 4.787 | −0.905 | 0.253 | 0.958 | −0.146 | −0.241 | 0.036 | 2.128 |
| 53 | 1.271 | −0.759 | 1.52 | −1.593 | 0.408 | | | | | |
| 55 | −3.453 | 3.621 | −1.876 | 3 | 0.574 | | | | | |
| 56 | | | 3.617 | 6.049 | 0.437 | 1.337 | | 4.27 | −0.197 | 6.637 |
| 57 | | 1.407 | −0.611 | 3.202 | 4.634 | | | | | |
| 58 | −0.927 | −1.168 | 1.95 | −0.431 | 0.902 | | | | | |
| 59 | | | −0.635 | | | | | | −0.98 | |
| 60 | | | | | −0.216 | | | | | |
| 61 | −0.031 | −0.436 | −0.32 | 2.243 | 3.074 | −0.509 | −1.327 | | 2.947 | −1.859 |
| 70 | −0.53 | 3.704 | 6.48 | −0.571 | 0.452 | | | | | |
| 73 | −1.149 | 0.603 | −0.382 | −2.614 | −2.861 | | | | 1.419 | |
| 74 | −0.604 | | −2.742 | 0.111 | −1.933 | 1.93 | −2.813 | −1.328 | 1.716 | −3.798 |
| 78 | 1.246 | 0.191 | −2.624 | −0.382 | 4.434 | | | | | |
| 79 | 1.603 | 1.497 | 0.101 | −1.166 | −0.518 | | | | | |
| 80 | | | | 4.342 | −0.277 | −0.755 | 0.165 | | | 0.515 |
| 81 | −0.585 | 0.597 | 1.048 | 0.888 | −3.466 | 2.316 | −1.876 | | | 2.449 |
| 83 | 4.763 | −3.23 | 3.327 | −0.53 | | 1.532 | 3.135 | −3.783 | 4.944 | −1.725 |
| 85 | 2.114 | −1.155 | 6.32 | −0.015 | 5.278 | −1.616 | −1.757 | 0.965 | 1.43 | 1.269 |
| 86 | 2.46 | −0.142 | 0.809 | −1.258 | 2.801 | | | | 0.434 | 3.327 |
| 87 | −0.204 | | 6.27 | −0.905 | 0.223 | | | | 3.797 | 6.117 |
| 88 | −0.68 | 1.302 | −0.573 | 1.453 | 0.378 | | 0.52 | 1.442 | −0.118 | 3.179 |
| 92 | | | 7.519 | −0.069 | 1.916 | | 0.071 | | 5.812 | −1.082 |
| 93 | | | −4.348 | 2.599 | 2.257 | | | | | |
| 95 | | | −2.184 | 1.057 | 2.03 | | | | | |

**Table A2.** *Cont.*

| Number | I | II | III | IV | V | I | II | III | IV | V |
|---|---|---|---|---|---|---|---|---|---|---|
| 96 | | −0.819 | | 5.815 | −0.313 | 1.468 | 0.385 | 4.601 | 3.225 | −3.157 |
| 97 | | | 1.693 | 3.789 | −4.912 | −0.313 | | −2.692 | 1.505 | 2.003 |
| 98 | | 2.962 | −1.056 | 2.413 | −3.321 | 0.987 | | −0.683 | 1.933 | 2.073 |
| 99 | | | | | 8.269 | | | | | 5.764 |
| 100 | | 0.129 | −0.039 | −2.485 | −0.273 | −0.609 | −0.778 | −0.231 | 1.684 | 1.267 |
| 101 | | | | −0.467 | 4.491 | | | | −3.885 | 1.429 |
| 102 | | | 5.946 | −2.458 | 0.569 | | 0.608 | | 10.275 | −1.74 |
| 103 | | | | | 6.342 | −0.322 | −1.59 | 1.585 | −1.504 | 2.641 |
| 104 | | | | | 1.788 | | | | | −1.157 |
| 105 | 1.063 | 1.039 | 7.541 | 1.282 | −2.503 | | | | | |
| 106 | | | | 3.602 | −0.802 | | | | 6.329 | −2.26 |
| 107 | −1.115 | | | 5.024 | −11.934 | | 1.614 | | | 0.492 |
| 108 | | | | 2.961 | 2.211 | | | −4.648 | 5.543 | −0.115 |
| 111 | | | | 9.034 | −7.233 | | | | 9.097 | −5.063 |
| 112 | | | 1.313 | 0.258 | 0.567 | 0.98 | | −2.345 | 4.857 | 0.752 |
| 113 | | | 0.808 | 6.757 | −11.817 | | | 6.047 | 1.03 | −7.838 |
| 115 | | | −0.004 | 5.786 | −2.76 | | | 2.376 | −0.453 | 0.542 |

| Number | NH$_3$-N(15–17) I | II | III | IV | V | NH$_3$-N(18–20) I | II | III | IV | V |
|---|---|---|---|---|---|---|---|---|---|---|
| 1 | 14.143 | | 1.101 | −0.113 | | 14.22 | | | | |
| 2 | 14.527 | −2.073 | 4.119 | 0.154 | −0.301 | 22.046 | | 1.265 | 0.68 | |
| 3 | 2.223 | 2.654 | | 1.113 | | | | | | |
| 4 | 9.104 | 0.842 | 4.047 | 0.981 | | 4.066 | −4.149 | | −0.664 | |
| 5 | 11.504 | 4.803 | −0.653 | 0.121 | | 0.166 | | | | |
| 6 | 13.132 | | 0.526 | | | 10.565 | | | | |
| 7 | −4.613 | | 0.602 | −0.852 | | 0.843 | | 0.682 | | |
| 8 | 1.003 | 4.308 | 0.844 | 0.334 | 3.319 | −2.398 | 3.559 | 0.144 | −0.552 | 1.073 |
| 10 | −0.518 | 1.58 | 0.842 | | | 7.258 | 3.774 | −0.075 | −0.11 | 0.893 |
| 11 | −6.686 | 2.773 | 0.687 | 1.001 | 1.87 | 11.67 | −2.414 | 1.395 | 0.076 | |
| 12 | 8.866 | −1.409 | 1.81 | −0.084 | 0.28 | 2.43 | 4.884 | −1.547 | 0.098 | |
| 13 | −3.012 | 1.62 | −0.296 | −0.314 | | −0.652 | 7.826 | | 1.23 | −1.935 |
| 14 | −0.293 | 2.238 | 4.507 | 1.845 | 1.054 | −15.745 | 3.705 | 1.93 | 0.692 | 0.584 |
| 15 | −15.948 | 5.054 | | | | −10.774 | 3.741 | 0.645 | 0.505 | |
| 16 | −3.415 | 3.334 | 3.167 | 0.007 | 1.944 | 2.588 | 6.745 | 1.894 | 2.215 | 0.047 |
| 17 | −0.775 | 2.657 | 1.39 | 1.94 | | −7.393 | | | −1.065 | |
| 18 | −7.831 | 2.76 | 2.686 | 0.016 | 1.532 | −25.245 | | −0.828 | 0.254 | −0.801 |
| 19 | 0.212 | 2.523 | 2.243 | −0.694 | 0.741 | −5.992 | 2.591 | 2.007 | −1.402 | −0.326 |
| 20 | 9.994 | | 0.488 | −0.616 | | 15.345 | 1.794 | | | 1.631 |
| 21 | −5.612 | 1.172 | 3.299 | 1.827 | 0.275 | 9.961 | | 5.28 | | |
| 22 | 3.969 | 1.974 | −0.461 | 1.081 | | 4.001 | 1.744 | | 1.634 | 1.353 |
| 23 | 13.762 | −5.056 | 1.388 | 1.123 | | 3.88 | | 0.116 | 0.931 | 0.274 |
| 24 | −12.106 | 4.603 | 3.108 | 0.114 | 0.161 | 1.944 | | | −1.735 | |
| 25 | −5.768 | 1.783 | −0.645 | | | 5.865 | 5.478 | 2.719 | 1.558 | 0.083 |
| 26 | 4.922 | 6.205 | 0.145 | | | 4.516 | 4.102 | −1.46 | −0.823 | |
| 27 | 1.403 | −5.868 | −1.723 | | | 8.572 | 5.997 | −0.146 | 0.657 | 0.957 |
| 28 | 4.768 | | 1.434 | 0.783 | 1.353 | 13.867 | 4.83 | −0.279 | −1.013 | |
| 29 | 1.207 | 0.508 | 1.656 | −0.721 | | −6.647 | 4.363 | 1.493 | −2.624 | 1.896 |
| 30 | 3.734 | −1.91 | 1.759 | 0.967 | | −7.549 | | −0.154 | 0.012 | −0.022 |
| 31 | 0.501 | | 0.012 | 0.388 | 0.704 | −15.483 | −9.073 | 0.453 | 0.279 | |
| 32 | 1.417 | −1.016 | −0.265 | −0.623 | −1.537 | 4.781 | 3.473 | −1.735 | 1.748 | 0.5 |
| 35 | 14.531 | | | | 0.532 | 12.275 | −13.431 | −0.282 | | −0.62 |
| 36 | 5.004 | 5.336 | −0.289 | −0.201 | | −7.392 | | 0.867 | | 0.977 |
| 37 | 3.024 | | 0.887 | 0.29 | | 0.173 | | | | |
| 38 | 2.727 | −2.128 | 0.053 | | 0.23 | | | | | |
| 39 | 7.571 | 9.21 | 1.707 | −0.137 | −1.611 | 13.3 | 1.866 | 1.1 | 2.675 | 1.486 |
| 41 | 12.904 | | 0.736 | | | 15.677 | | 0.839 | 0.36 | |
| 42 | −6.365 | | | | | −23.257 | 0.124 | −1.245 | 0.033 | |
| 43 | −18.962 | 0.6 | 1.102 | | | −13.794 | 2.774 | −1.686 | 0.866 | 0.195 |
| 46 | −3.607 | | 2.585 | | | −4.392 | 3.119 | 0.487 | 1.034 | 1.252 |
| 47 | 2.621 | | 4.2 | −0.961 | | −1.518 | 1.74 | −0.814 | 1.696 | −0.449 |
| 48 | 2.02 | −4.334 | 6.285 | −0.964 | −1.963 | −16.288 | −0.495 | 0.215 | 0.843 | 0.777 |
| 53 | 16.207 | | −0.858 | | | | | | | |
| 55 | 2.343 | | | | | | | | | |
| 56 | 19.113 | | −1.223 | | | 17.904 | | | | |
| 57 | −0.98 | | | | | | | | | |
| 58 | 11.213 | 3.202 | −1.443 | 0.971 | | | −0.138 | | | |

**Table A2.** *Cont.*

|  |  |  |  |  |  |  |  |  |  |  |
|---|---|---|---|---|---|---|---|---|---|---|
| 59 |  |  |  |  |  |  |  |  |  |  |
| 60 | 3.695 |  |  |  |  |  |  |  |  |  |
| 61 | 7.63 |  |  |  | −0.132 | 11.57 |  |  |  |  |
| 70 | 6.073 | −3.497 | 0.813 |  | 0.229 |  |  |  |  |  |
| 73 | 7.916 | −4.129 | 4.839 | −1.51 | 0.701 |  |  |  |  |  |
| 74 | 11.385 | −2.785 | −3.407 | 0.937 | −0.571 | 17.349 |  | −0.386 | 0.042 |  |
| 78 | 11.204 |  | 1.004 |  |  |  |  |  |  |  |
| 79 | −1.094 | 0.172 | 1.319 | 1.693 | 1.109 |  |  |  |  |  |
| 80 | 4.668 | 3.023 | 1.344 | 0.351 |  | 0.252 | 3.87 | 1.52 | 1.397 | −0.218 |
| 81 | −4.245 | 2.488 |  | −0.635 |  | 1.238 | 1.948 | −0.473 | −1.843 |  |
| 83 | −3.394 | 1.705 | 1.453 | −0.949 | 1.185 | 12.455 | 3.56 | 0.367 | 1.857 | −0.545 |
| 85 | 2.242 | 3.146 | 5.71 | 2.61 | 0.423 | 1.772 | 0.707 | −0.071 | −1.916 | 0.837 |
| 86 | −1.064 |  | 1.072 | 1.044 | −0.374 | 17.563 |  | 1.697 | 0.15 |  |
| 87 | 20.281 |  | −1.013 |  |  | 16.09 |  | −1.403 |  |  |
| 88 | 8.378 | 4.705 | −0.096 | 0.019 |  | 6.037 | 3.502 | 1.351 | 0.085 | 0.967 |
| 92 | 15.208 |  | −0.38 | −0.422 |  | 10.407 |  | −0.155 |  |  |
| 93 | 3.937 |  |  |  | 0.595 |  | −0.646 |  |  |  |
| 95 | 10.428 |  | −0.077 |  |  |  |  |  |  |  |
| 96 | 13.843 | −0.335 | 0.863 | 0.577 |  | 11.329 | −3.804 |  | 1.072 | −0.233 |
| 97 | −13.736 |  |  |  |  | −19.046 |  | −0.233 |  | 0.757 |
| 98 | 3.531 | 3.312 |  | −1.495 | 0.629 |  | −5.172 | 4.018 | −0.673 | 1.655 |
| 99 | −12.274 |  |  |  |  | −16.997 | −1.304 |  |  |  |
| 100 | −14.906 |  |  |  | −0.229 | −26.228 | 0.491 | 0.458 |  |  |
| 101 | 3.622 |  | 0.199 |  |  | 5.575 |  |  |  |  |
| 102 | 13.517 |  | 1.376 | 0.156 | 1.326 | 2.133 |  | 0.371 |  |  |
| 103 | 15.781 |  |  |  |  | −7.436 |  | 2.053 | −0.647 |  |
| 104 | −16.504 |  |  |  |  | 0.225 |  |  |  |  |
| 105 | 3.193 | 1.44 | 0.011 |  | −0.213 |  |  |  |  |  |
| 106 | 4.998 |  |  | −0.573 |  | 2.617 | 2.998 | 0.917 |  |  |
| 107 | 14.963 |  |  |  |  | 8.314 |  |  |  |  |
| 108 | −6.62 |  | −0.241 | −0.779 |  | −7.654 | 0.229 | 0.51 |  |  |
| 111 | 5.13 |  | −0.979 | 0.118 | −0.605 | −1.831 |  | −0.557 |  |  |
| 112 | −3.757 |  | 1.394 | −0.169 |  | −13.988 |  |  | −1.316 |  |
| 113 | −2.155 |  |  | 0.733 |  | −19.447 |  | −0.406 |  |  |
| 115 | 13.721 |  |  |  |  | −10.261 |  | 2.12 | 1.48 | 0.202 |

Note: Blank spaces occur where there was not insufficient sample size near the classification points of the cut-offs to provide a result.

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
