# Peer review of "Has Third-Party Monitoring Improved Water Pollution Data Quality? Evidence from National Surface Water Assessment Sections in China"

_water, doi:10.3390/w13202917_

Round 1

Reviewer 1 Report

Repetition of the same things concerning the effectiveness of third-party monitoring in water environment monitoring and its gains.

The three elements monitored are certainly not enough to classify the water quality.

In the figures it must be clear what each axis is about.

In Table 3 the Standard limits of basic items of surface water quality do not differ among the five classes. It is strange.

In Table 4. The mean and std dev of the descriptive statistics of the sample don’t differ much.  It is strange.

Author Response

The specific modification content is shown in word

Reviewer 2 Report

1. In the Introduction, there is no references to the situation in other countries of the world. Do the Authors have such information? Please search and refer to them.
2. Please explain in the Method, why the McCrary method was used? Have other methods been tried? What are the limitations of the McCrary method? What is the biggest weakness of the regression discontinuity approach?
3. What is manipulation of the running variable?
4. Incorrect references in the text. Eg: line 156: Jingdong (2012) proposed ... "It should be:" Jingdong in work [...] proposed ... ". Please include such changes throughout the all text.
5. Check punctuation marks, pauses are missing in many places.
6. In Figure 3, the X and Y axis signatures are missing.
7. In figure 4, please describe the curve equation.
8. Please explain the different values of the number n in Figure 4.

Author Response

The specific modification content is shown in Word

Round 2

Reviewer 1 Report

Compared with the air quality index, the water quality index assessment and supervision process are more complicated, as you have mentioned. It is rather dangerous to be based only on physico-chemical indicators to evaluate the water quality; the ecological quality is very important. This is why in USA and Europe they have included hydromorphological and biological indicators.

The three elements of monitoring are indeed not enough to classify water quality, and as you mention at the current research results are based only on a very short-term impact and the impact on water quality is not so obvious leading to a possibility the improvement of the water quality not to be adequate in the future. Additionally, you do not comment in your discussion the need to include other important indicators in the on-site monitoring, as I mentioned above.

Although you have shown that the third-party monitoring method can reduce data manipulation, improve the quality of water environment information disclosure, this cannot expand the field of existing research because there is not foresight according to the bidding documents of the National Surface Water Environmental Monitoring Network for manual monitoring of cross-section monitoring technical services issued by the China Environmental Monitoring Center, the on-site monitoring items to include hydromorphological and biological indicators.
